# The circadian clock influences T cell responses to vaccination by regulating dendritic cell antigen processing

Mariana P. Cervantes-Silva[1,9], Richard G. Carroll [1,9], Mieszko M. Wilk [2,3], Diana Moreira[2], Cloe A. Payet[1], James R. O'Siorain[1], Shannon L. Cox[1], Lauren E. Fagan[1,4], Paula A. Klavina [1,5], Yan He[1,6], Tabea Drewinski[1], Alan McGinley[1], Sharleen M. Buel[7], George A. Timmons[1], James O. Early[1,4], Roger J. S. Preston[5], Jennifer M. Hurley[7], David K. Finlay [2], Ingmar Schoen [5], F. Javier Sánchez-García[8], Kingston H. G. Mills[2] & Annie M. Curtis [1,2,4,5] ✉

Dendritic cells play a key role in processing and presenting antigens to naïve T cells to prime adaptive immunity. Circadian rhythms are known to regulate many aspects of immunity; however, the role of circadian rhythms in dendritic cell function is still unclear. Here, we show greater T cell responses when mice are immunised in the middle of their rest versus their active phase. We find a circadian rhythm in antigen processing that correlates with rhythms in both mitochondrial morphology and metabolism, dependent on the molecular clock gene, *Bmal1*. Using Mdivi-1, a compound that promotes mitochondrial fusion, we are able to rescue the circadian deficit in antigen processing and mechanistically link mitochondrial morphology and antigen processing. Furthermore, we find that circadian changes in mitochondrial $Ca^{2+}$ are central to the circadian regulation of antigen processing. Our results indicate that rhythmic changes in mitochondrial calcium, which are associated with changes in mitochondrial morphology, regulate antigen processing.

Dendritic cells (DCs) phagocytose, process and present antigens to naive T cells and thereby play a critical role in priming adaptive immune responses to infection or following vaccination. Cells of the innate and adaptive immune system, including DCs, express all the components of the endogenous molecular clock and display circadian rhythmicity in gene expression[1]. These cellular timers produce daily oscillations in a range of critical immune cell functions, such as phagocytosis[2], cytokine production[3], cell trafficking[4], cell migration[5] along with anti-parasite[6], antibacterial and antiviral immune responses[5,7]. In elderly individuals, vaccination against influenza in the morning generates higher antibody titres than vaccination in the afternoon[8]. Individuals immunised with the BCG vaccine in the morning showed stronger trained immune responses compared with those vaccinated in the evening[9]. However, the mechanisms modulating these time-of-day differences in immune responses to human vaccines remain poorly understood[8].

[1]Curtis Clock Laboratory, School of Pharmacy and Biomolecular Sciences, Royal College of Surgeons in Ireland RCSI, Dublin, Ireland. [2]School of Biochemistry and Immunology, Trinity Biomedical Sciences Institute, Trinity College Dublin, Dublin, Ireland. [3]Department of Immunology, Faculty of Biochemistry, Biophysics and Biotechnology, Jagiellonian University, Krakow, Poland. [4]Tissue Engineering Research Group (TERG), Royal College of Surgeons in Ireland RCSI, Dublin, Ireland. [5]Irish Centre for Vascular Biology, School of Pharmacy and Biomolecular Sciences, Royal College of Surgeons in Ireland RCSI, Dublin, Ireland. [6]Institute of Functional Nano & Soft Materials (FUNSOM), Jiangsu Key Laboratory for Carbon-Based Functional Materials and Devices, Soochow University, Suzhou, China. [7]Department of Biological Sciences & Center for Biotechnology and Interdisciplinary Sciences, Rensselaer Polytechnic Institute, Troy, NY 12180, USA. [8]Immunoregulation Laboratory, Department of Immunology, Escuela Nacional de Ciencias Biológicas, Instituto Politécnico Nacional, México City, Mexico. [9]These authors contributed equally: Mariana P. Cervantes-Silva, Richard G. Carroll. ✉e-mail: anniecurtis@rcsi.com

Metabolic changes in DCs can influence their differentiation and activation[10–12]. Mitochondria are central to cellular metabolic changes and are a major site of energy production in the form of ATP. Mitochondria are highly dynamic organelles in terms of their spatial distribution, morphology and inner architecture and change their shape continually to meet the energetic demands of the cell. Cells can alter their mitochondrial morphology through altering the balance of fusion and fission processes within the cell. Mitochondrial fusion results in the joining of two mitochondria into one and is mediated by MFN proteins and the GTPase OPA1[13]. While mitochondrial fission involves the division of a single mitochondrion into two daughter mitochondria and is mediated in part by the cytosolic GTPase DRP1[13]. Recently, we reported that mitochondrial dynamics in mouse macrophages are tightly controlled by the molecular clock. Mitochondria displayed enhanced fusion during the rest phase and fission during the active phase. The fused state at the end of the resting phase correlated with higher ATP production and phagocytic function[14]. These oscillations in oxidative phosphorylation (OXPHOS) and ATP generation have been demonstrated across a range of cell types, and are associated with the NAMPT-NAD-SIRT1/3 axis and dependent on circadian modification of DRP1 phosphorylation[15–17]. However, it is still unknown whether mitochondrial morphology is under clock control in DCs and what impact this might have on DC function.

Mitochondria also act as a calcium sink and regulate the levels of cytosolic calcium and modulate biological pathways. It has been shown that the molecular clock drives oscillations in mitochondrial calcium and ATP production in astrocytes[18]. Calcium is crucial for chemokine-dependent migration of DCs. Engagement of chemokine receptors, such as CCR7, triggers trafficking of DCs by increasing $Ca^{2+}$ influx[19–21]. Interestingly, Holtkamp et al. demonstrated that skin DCs preferentially migrate into lymphatic vessels during the mouse rest phase due to clock control of CCR7[22]. Thus, control of cellular $Ca^{2+}$ is an important determinant of DC function, but the molecular mechanisms regulating $Ca^{2+}$ localisation within DCs and its consequences are not well understood.

Therefore, we sought to investigate the impact of circadian rhythms on DC function and whether this might explain the diurnal variability observed in the protective immune response induced by certain vaccines. There is a growing body of evidence to suggest that cellular metabolism and mitochondria are central to DC function[23,24]. However, it is still unclear whether circadian rhythms might intersect with these metabolic pathways to impart time-of-day influences on DC function. In this study, we found that the DC molecular clock controls calcium mobilisation, mitochondrial dynamics and metabolism to influence antigen processing and T cell activation. Understanding these functionally important daily oscillations of DCs may determine a time-of-day that is optimal for vaccination, and may also uncover useful approaches for boosting DC function, to enhance vaccine-induced immunes responses.

## Results

### DCs play a critical role in the time-of-day T cell response to immunisation

Circadian rhythms regulate many aspects of innate immunity[25,26]. Response to vaccination is time-of-day dependent but the underlying mechanisms are unclear[8,9,27,28]. A recent study demonstrated rhythms in T cell proliferation following adoptive transfer of antigen loaded BMDCs in mice[29]. In this study, DCs were pre-loaded with OVA peptide in vitro. Peptides can bypass intracellular processing pathways and directly associate with MHCI/II. Therefore, that experimental design did not assess the effect of circadian rhythms on antigen uptake and processing by DCs and as such prompted us to investigate the impact of the clock on antigen uptake and processing in DCs. We utilised an adoptive transfer model (Fig. 1a), where CD4⁺ T cells were isolated from OT-II mice on Day −1 at ZT3 and stained with cell trace violet

(CTV). These CTV-stained OTII CD4⁺ T cells were immediately injected into recipient mice that had been phase shifted to either ZT7 or ZT19 in light cabinets to facilitate simultaneous experimentation. The next day ZT7 and ZT19 mice were immunised with OVA and whole cell pertussis (wcP) as an adjuvant, and 72 h later (again corresponding to either ZT7 or ZT19), mediastinal lymph nodes were harvested from these mice to analyse proliferation and activation of the CTV-stained T cells. We reasoned that the approach of injecting the same CTV-stained OTII CD4⁺ T cells into phase-shifted recipient mice allowed us to more accurately interrogate the effect of the DC molecular clock on T cell activation (Fig. 1a and Supplementary Fig. 1a).

Zeitgeber time (ZT) is defined as the time in hours following the onset of light in the animal facility. Mice were maintained in a 12 h:12 h light:dark environment. For example, ZT7 refers to 7 h after lights on and the middle of the rest phase, ZT19 refers to 7 h after lights off and the middle of the active phase.

Flow cytometry analysis of the CTV-stained OTII CD4⁺ T cells revealed a significant increase in T cell proliferation in the mediastinal lymph node when mice were immunised at ZT7 versus ZT19 (Fig. 1b–d), with no T cell proliferation observed in PBS controls. There were more CTV⁺ OTII T cells in the mediastinal lymph node in mice that were immunised at ZT 7 compared to ZT 19 (1.7% versus 0.8%) (Fig. 1c). Using CD69 as a marker of activated T cells, we found more activated CTV⁺ OTII T cells in the mediastinal lymph nodes of mice immunised at ZT7 versus ZT19 (Fig. 1e, f). These results suggest that molecular processes within the DC required for T cell activation must be under control of the molecular clock. Thus, we decided to further investigate the molecular mechanisms underpinning this effect.

### Synchronised DCs display robust rhythms in clock gene expression and antigen processing

Antigen uptake (phagocytosis), processing and presentation are essential for DC activation of T cells. Therefore, we investigated which of these, if any, might be under the control of the molecular clock and provide an explanation for the time-of-day vaccine effect observed in Fig. 1. Firstly, we investigated whether DCs contained a functional clock with circadian rhythms. Cells can be synchronised in vitro to the same circadian phase through the addition of serum rich media (50% horse serum) for 2 h followed by serum withdrawal (serum-shock)[30] (Fig. 2a). Following synchronisation, PER2::Luc bone marrow-derived dendritic cells (BMDCs) produced strong circadian rhythms as demonstrated by oscillations of luminescence resulting from the cycling of PER2 protein that persisted up to 4–5 days in culture (Fig. 2b). Analysis of expression of the molecular clock genes, *Period 2*, *Nr1d1* and *Bmal1* by qPCR revealed that they were also cycling (Supplementary Fig. 2a).

We next investigated the regulation of antigen processing in DCs by the molecular clock using DQ-OVA. DQ-OVA is a self-quenched labelled conjugate of the full length ovalbumin protein, which produces a fluorescent green signal following cleavage into peptide fragments[31]. *Bmal1* is a core clock gene forming part of the positive arm[32]. Deletion of *Bmal1* ablates the oscillation of the proteins in the negative arm of the core clock and cannot be compensated by any other paralogues in the native context. Thus, we used *Bmal1*-deleted BMDCs as a model of circadian disruption. Antigen processing was quantified using DQ-OVA in synchronised *Bmal1*⁺/⁺ and *Bmal1*⁻/⁻ BMDCs at distinct timepoints across the circadian cycle up to 48 h. We found that antigen processing in *Bmal1*⁺/⁺ BMDCs followed a circadian rhythm, with highest processing at 12 h and 36 h and the lowest at 24 h and 48 h post synchronisation (Fig. 2c, d). In contrast, BMDCs lacking *Bmal1* had low levels of antigen processing and lacked any discernible rhythm at all times tested (Fig. 2c, d). These findings demonstrate a significant role for the molecular clock in regulation of antigen processing. Differences in antigen uptake, measured by FITC-OVA, did not differ by time-of-day or by genotype (Supplementary Fig. 2c), suggesting specific clock controlled regulation on antigen processing.

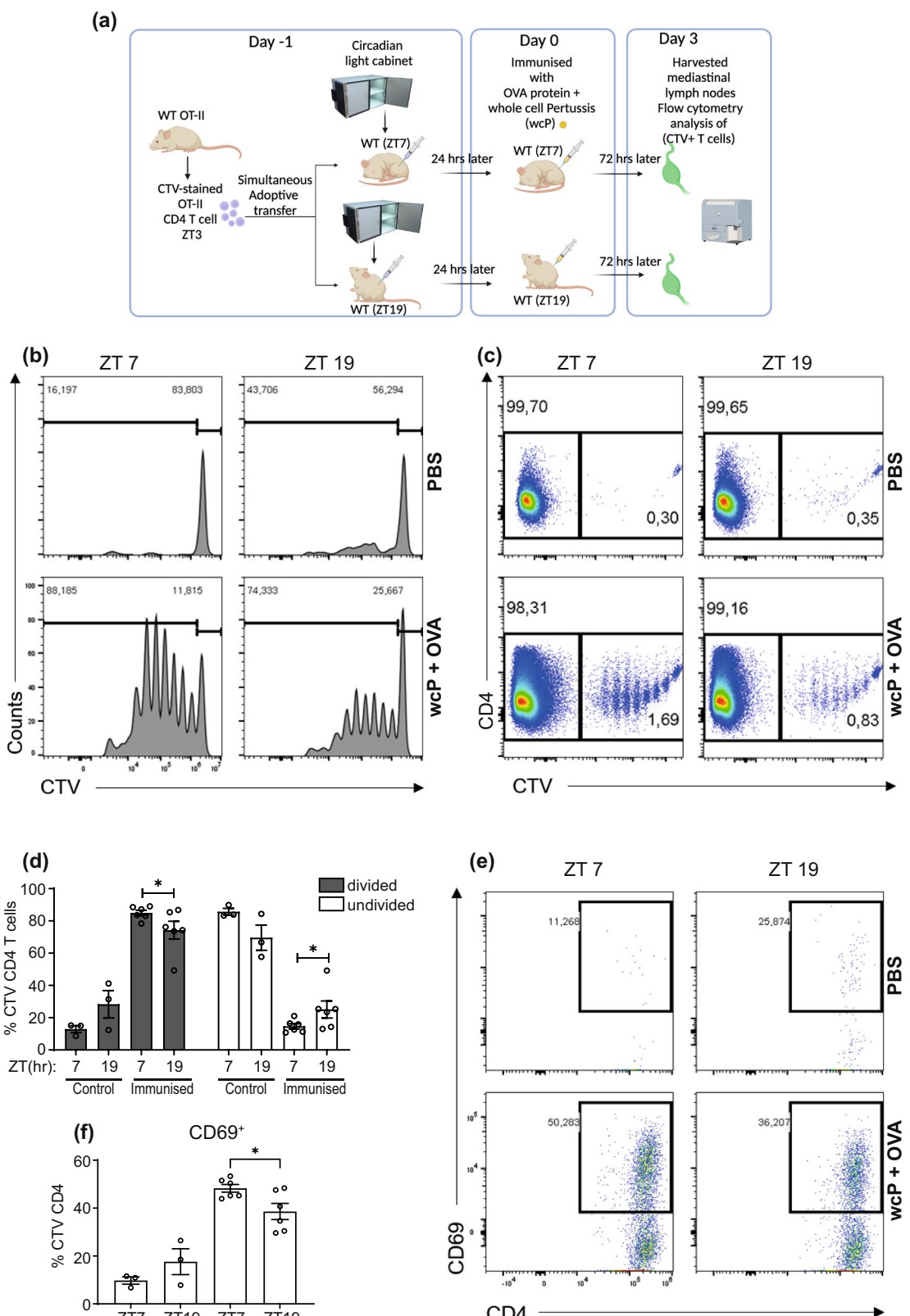

We next tested whether the circadian variation of antigen processing was present in primary antigen presenting cells (APCs). To achieve this, splenic cells were isolated from mice at ZT1, ZT7, ZT13 and ZT19 and antigen processing was measured by DQ-OVA fluorescence.

Analysis of the CD11b and CD11c populations revealed that the percentage of cells processing DQ-OVA increased from 55% at ZT7 to 75% at ZT19 in CD11b⁺, and from 28% at ZT7 to 43% at ZT19 in CD11c⁺ cells (Fig. 2e). We also observed that antigen processing was reduced in CD11b⁺ cells within the spleen of *Bmal1^{myeloid−/−}* mice when compared with *Bmal1^{myeloid+/+}* mice even though they had similar number of CD11b⁺ cells (Fig. 2f). In the total DC population, we found approximately 2-fold increase in processing of DQ-OVA from ZT7 and ZT19 (Fig. 2g

**Fig. 1 | T cell activation and proliferation is dependent on time-of-day of immunisation. a** Experimental design for adoptive transfer of labelled T cells and immunisations by circadian phase. CTV⁺ OT-II CD4⁺ T cells (harvested at ZT3) were transferred directly into ZT7 or ZT19 recipient mice. 24 h later immunisations of ZT7 or ZT19 recipient mice occurred. Immunisations were performed using wcP vaccine + OVA 10 µg/mouse ($n$ = 6 mice) or PBS control ($n$ = 3 mice) and mediastinal lymph nodes harvested 72 h later. **b, c** Proliferation of CTV⁺ stained OT-II CD4⁺ T cells harvested from mediastinal lymph node. **d** Percentage of divided and undivided CTV⁺ stained OT-II CD4⁺ T cells ($n$ = 6 immunised mice or $n$ = 3 control mice) $p$ = 0.04 for divided and undivided cells. **e** Representative plot of CD69⁺ expression on CTV⁺ stained OT-II CD4⁺ T cells. **f** Percentage of CD69⁺ expression on CTV⁺ stained OT-II CD4⁺ T cells ($n$ = 6 immunised mice or $n$ = 3 control mice) $p$ = 0.02. Data shown is mean with error bars representing ± SEM. Data were compared using two-tailed $t$-test, *$p$ < 0.05. Source data are provided as a Source Data file.

and Supplementary Fig. 3a). We next investigated different sub-populations, including cDC1s, which are associated with cross presentation and activation of MHCI, cDC2s which are associated with conventional MHCII presentation and plasmacytoid DCs that are prominent in viral infection (Supplementary Fig. 1b)[33]. The cDC1 subpopulation increased by over 2-fold (ZT19 – 47.7% vs ZT7 – 20.8%) and the cDC2 population also increasing at ZT19 (62%) compared to ZT7 (44%) (Fig. 2g and Supplementary Fig. 3a). Plasmacytoid DCs and macrophages also showed a similar time-of-day dependency with higher antigen processing at ZT19 compared to ZT7 (Fig. 2g and Supplementary Fig. 3b). Both resident and migratory cells also displayed higher antigen processing at ZT 19 as measured by DQ-OVA processing (Fig. 2h and Supplementary Fig. 3c). The resident cDC are the major DC population in the spleen, and execute their antigen collection, processing and presentation within that lymphoid organ. The migratory cDCs constantly sample the tissue, and once activated, travel to the draining lymph node where they encounter naive T cells[34]. These ex vivo results demonstrate a clear time-of-day difference in the antigen processing function across all DC subtypes and macrophages.

### DC mitochondrial metabolism is regulated by the molecular clock and is required for antigen processing

Metabolism is a key regulator of DC function, and the molecular clock controls cellular metabolism[12,24,35]. We next investigated if DC metabolism varied in a circadian manner and whether this influenced the observed rhythms in antigen processing. We compared mitochondrial metabolism between *Bmal1*⁺/⁺ and *Bmal1*⁻/⁻ BMDCs at distinct time points post serum synchronisation using the Agilent Seahorse XF Cell Mito Stress Test. We found that mitochondrial metabolism in *Bmal1*⁺/⁺ BMDC displayed a rhythmic phenotype, with highest maximal respiration, spare respiratory capacity (SRC) and ATP levels at 12 h and 36 h post synchronisation, and lowest at 24 h post synchronisation (Fig. 3a–d). In contrast, mitochondrial metabolism did not show any rhythmicity in these readouts in *Bmal1*⁻/⁻ BMDC and were consistently lower when compared to *Bmal1*⁺/⁺ BMDC (Fig. 3a–d). These results demonstrate that mitochondrial metabolism is regulated by the endogenous molecular clock in DCs. To investigate whether mitochondrial metabolism was involved in antigen processing by DCs, the mitochondrial metabolism inhibitors Oligomycin (an inhibitor of ATP synthase) and FCCP (a mitochondrial oxidative phosphorylation uncoupler) were used in the DQ-OVA antigen-processing assay at 12 h post synchronisation. Oligomycin and FCCP significantly reduced antigen processing in both *Bmal1*⁺/⁺ and *Bmal1*⁻/⁻ BMDCs (Fig. 3e, f), indicating the importance of mitochondrial metabolism for antigen processing. As expected both inhibitors significantly reduced ATP levels. (Supplementary Fig. 4a). To investigate if the reduced antigen processing conferred a biological effect on T cells, coculture experiments were performed with BMDCs and OVA specific OTII CD4⁺ T cells in the presence or absence of mitochondrial metabolism inhibitors. To rule out any effects of the inhibitors on T cells, BMDCs were pretreated with mitochondrial inhibitors, followed by OVA and then BMDCs were washed thoroughly before T cells were added. Addition of OVA to the cocultures OTII CD4⁺ T cells and BMDCs resulted in robust production of IFN-γ and IL-17 from activated T cells that was significantly inhibited with the addition of oligomycin or FCCP

(Fig. 3g). The reduced antigen processing observed in *Bmal1*⁻/⁻ compared with *Bmal1*⁺/⁺ BMDCs also resulted in significantly lower IFN-γ production by OTII CD4⁺ T cells (Fig. 3h). Collectively, these results identify mitochondrial metabolism as a key regulator of antigen processing in DCs and reveal that DC molecular clock regulation of mitochondrial metabolism modulates antigen processing function and subsequent T cell activation.

### DC mitochondrial morphology and mitochondrial potential display circadian rhythmicity

As mitochondrial morphology is known to directly influence mitochondrial metabolism[36,37], we investigated if the observed circadian oscillations in antigen processing were accompanied by alterations in the mitochondrial morphology network of DCs. We observed clear circadian rhythms in mitochondrial morphology within DCs that were dependent on *Bmal1*. In *Bmal1*⁺/⁺ BMDCs, the mitochondria displayed a fragmented phenotype at 24 h and 48 h post serum synchronisation (Fig. 4a), which correlated with times when antigen processing was lowest (Fig. 2c). In contrast, mitochondria had a fused phenotype at 12 and 36 h post serum synchronisation in *Bmal1*⁺/⁺ BMDCs (Fig. 4a, c, d), correlating with high levels of antigen processing (Fig. 2c). In *Bmal1*⁻/⁻ BMDCs, mitochondria were predominantly fragmented at all time-points tested (Fig. 4b–d) correlating with low antigen processing (Fig. 2c). These observed changes in mitochondrial fragmentation displayed significant circadian rhythmicity when analysed by cosinor analysis (Fig. 4e), although, the mitochondrial fusion failed to reach significance for circadian rhythmicity (Fig. 4f). Mitochondrial membrane potential is a key indicator of mitochondrial metabolism[37,38], and this also displayed a circadian rhythm by cosinor analysis (Fig. 4g). These circadian changes in mitochondrial potential were not due to changes in mitochondrial mass; analysis of mitochondrial mass using mitotracker green showed no time-of-day or *Bmal1* dependent variation (Fig. 4h). We also investigated mRNA and protein expression of key genes involved in mitochondrial morphology, such as *Fis1, Opa1, Mfn1* and *Mfn2*. None of these genes displayed any circadian variation at mRNA level (Supplementary Fig. 5a), however the mitochondrial fission gene *Fis1* was rhythmic at protein level in synchronised *Bmal1*⁺/⁺ with an increase at 24 h post serum synchronisation but no rhythms were observed in *Bmal1*⁻/⁻ BMDCs (Supplementary Fig. 5b, c). Together, these results reveal a role for the DC molecular clock in the circadian regulation of mitochondrial metabolism, morphology and accompanying circadian rhythm of DC function.

### Control of mitochondrial morphology with small molecule inhibitors of DRP1 can alter the circadian-dependent antigen presenting function of DCs

Up to this point, we observed circadian rhythms in mitochondrial metabolism and morphology that correlated with antigen processing in DCs. We next wished to investigate the mechanistic link between circadian controlled mitochondria morphology and DC antigen processing. Firstly, we employed the small molecule Mdivi-1, an inhibitor of the pro-fission protein DRP1[39], to increase mitochondrial fusion at all timepoints following synchronisation. In *Bmal1*⁺/⁺ BMDC, Mdivi-1 treatment promoted mitochondrial fusion at 24 h post synchronisation (Fig. 5a), the time in the circadian cycle when mitochondria were

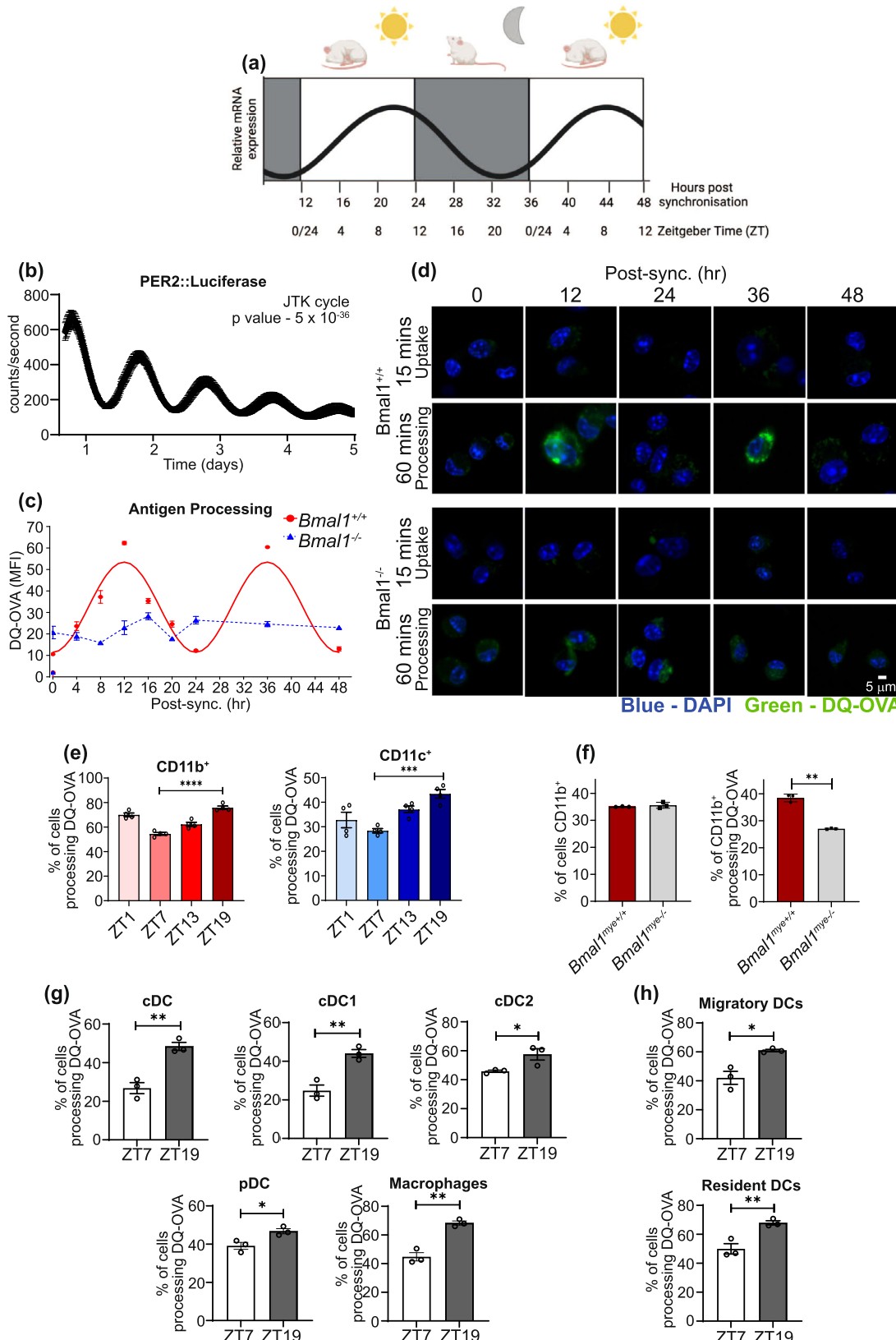

in a fragmented phenotype (Fig. 2e). However, there was no observable increase in fusion at 12 h post synchronisation in *Bmal1*[+/+] BMDC (Fig. 5a) as the mitochondria at this time point already display increased fusion under normal rhythmic control (Fig. 4a, c). Mitochondria displayed a fragmented phenotype in *Bmal1*[-/-] BMDCs at all timepoints post synchronisation and application of Mdivi-1 promoted

mitochondrial fusion (Fig. 5a). Critically, Mdivi-1 not only promoted mitochondrial fusion but also promoted antigen processing. The low level of antigen processing in *Bmal1*[+/+] BMDCs at 24 h post synchronisation was significantly boosted with addition of Mdivi-1 (Fig. 5b, d). Importantly, Mdivi-1 treatment could rescue the defective antigen processing that was consistently observed in *Bmal1*[-/-] BMDCs at 12 and

**Fig. 2 | Dendritic cells display robust circadian rhythms in antigen processing. a** A schematic summarising how ZT time can be inferred from the in vitro synchronised serum shock model comparing Per2 mRNA oscillation in BMDCs. Shading represents the relative active periods. As mice are nocturnal, 12 h post synchronisation represents ZT0 the onset of the inactive phase, whereas 24 h post synchronisation represents ZT12 the onset of the active phase. **b** Per2::luciferase BMDCs were synchronised by serum shock and circadian rhythms were measured using lumicycle technology (n = 3 biologically independent samples). *Bmal1$^{+/+}$* and *Bmal1$^{-/-}$* BMDCs were synchronised and antigen processing was measured at **c** 4 h or **d** 12 h intervals over a 48 h time course (n = 3 independent experiments). Antigen processing was measured by addition of DQ-OVA (1 μg/mL) and fluorescence (blue – DAPI, green – DQ-OVA) was measured at 15 min (uptake) or 60 min (processing) and then fixed and analysed by confocal microscopy. **e** Spleens were isolated from WT mice at ZT1, ZT7, ZT13 and ZT19 and single cell suspension generated, stained for DQ-OVA (1 μg/mL) as in (**c**, **d**) and subsequently stained for CD11b+ and CD11c+ and analysed by flow cytometry (n = 4 mice). **f** Spleens isolated from *Bmal1$^{myeloid+/+}$* and *Bmal1$^{myeloid-/-}$* mice and stained for DQ-OVA as in (**c**, **d**) and CD11b$^+$ and analysed by flow cytometry (n = 3 mice) p = 0.0045. **g, h** Splenic DCs were expanded by B16-FLT3L cells. **g** cDCs, cDC1s, cDC2s, plasmacytoid DCs and macrophages, or **h** migratory and resident DCs were identified by flow cytometry and DQ-OVA processing quantified by flow cytometry (n = 3–4 mice). cDC p = 0.003, cDC1 p = 0.005, cDC2 p = 0.04, pDC p = 0.02, macs p = 0.001, migratory DCs p = 0.01, resident DCs p = 0.0092 Data shown is mean with error bars representing ± SEM. Luciferase data was analysed for circadian rhythmicity by JTK cycle (**b**). Antigen processing in *Bmal1$^{+/+}$* was predicted to be circadian by cosinor analysis (**c**). Data were compared by one-way ANOVA with Tukey's post-hoc test for multiple comparisons (**e**) or by a two-tailed *t*-test (**f–h**). *p < 0.05, **p < 0.01, ***p < 0.001 and ****p < 0.0001. Source data are provided as a Source Data file.

24 h post synchronisation and Mdivi-1 treated *Bmal1$^{-/-}$* BMDCs had antigen processing levels comparable to *Bmal1$^{+/+}$* BMDCs (Fig. 5b–d). These results demonstrate that circadian oscillations in mitochondrial morphology are mechanistically linked to DC antigen processing.

## Circadian rhythms drive oscillations in mitochondrial calcium levels to coordinate antigen processing

The ability of mitochondria to act as a calcium sink has been reported to decrease cytosolic calcium concentrations, while also increasing mitochondrial ATP production[18]. Changes in cytosolic calcium concentrations will impact on calcium regulated enzymes, one of which is calcineurin, an enzyme that also drives mitochondrial fission through its effects on DRP1 phosphorylation[40,41]. We thus asked if calcium or calcium regulated enzymes influenced circadian controlled antigen processing in DCs. Firstly, we investigated if there was a time-of-day difference in the calcium localisation within the cell between the cytosol and mitochondria. Using Rhod-2, a mitochondrial calcium indicator, we observed that *Bmal1$^{+/+}$* BMDCs showed nearly 3-fold more calcium localised to the mitochondria at 12 h, relative to 24 h, post synchronisation (Fig. 6a, b). The opposite was true for cytosolic calcium levels assessed by Fluo-4 staining where we observed nearly 3-fold less cytosolic calcium at 12 h relative to 24 h post serum synchronisation (Fig. 6a–c). *Bmal1$^{-/-}$* BMDCs showed no time-of-day difference in the location of their calcium with low levels of mitochondrial calcium and high cytosolic calcium at both time points tested (Fig. 6a–c). This demonstrates that changes in calcium localisation are circadian and it suggests that the molecular clock may be using mitochondria as a calcium sink to control this temporal localisation. Mdivi-1, a compound that increases mitochondrial fusion, also increased mitochondrial calcium levels in both *Bmal1$^{+/+}$ and Bmal1$^{-/-}$* (Fig. 6d). We used a second molecule to increase mitochondrial fusion, a selective peptide P110 which promotes fusion by inhibiting DRP1 GTPase activity and the DRP1/FIS1 interaction[42]. We found that while P110 promotes mitochondrial fusion in *Bmal1$^{+/+}$*, it did not promote antigen processing to the same extent as Mdivi-1 (Supplementary Fig. 6a). Concurrently, we found that P110 did not increase mitochondrial calcium uptake (Supplementary Fig. 6b). Therefore, we reasoned that Mdivi-1 has such pronounced effects on antigen processing as it promotes both fusion and mitochondrial calcium uptake.

As calcineurin is a key regulator of mitochondrial morphology and is itself regulated by cytosolic calcium levels, we wished to investigate if calcineurin was mechanistically involved. Inhibition of calcineurin using FK506 was able to promote mitochondrial fusion and reduce mitochondrial fission at 24 h post synchronisation in both *Bmal1$^{+/+}$* and *Bmal1$^{-/-}$* genotypes (Fig. 6e). Importantly, FK506 was also able to boost the deficit in antigen processing in *Bmal1$^{+/+}$* BMDCs observed at 24 h post serum synchronisation (Fig. 6f). FK506 had no effect on antigen processing at 12 h or 36 h post serum synchronisation in *Bmal1$^{+/+}$* as mitochondria were in a

naturally fused state at this time-of-day due their circadian regulation (Fig. 6f). Similarly to Mdivi-1, FK506 was able to rescue the defect in antigen processing observed at all timepoints post synchronisation in *Bmal1$^{-/-}$* BMDCs (Fig. 6f). We hypothesised that the circadian control of antigen processing via mitochondria could be through ATP (Fig. 3d). However, neither Mdivi-1 or FK506 was able to enhance ATP levels (Supplementary Fig. 4b). Nonetheless, these results demonstrate a previously uncharacterised role for the endogenous molecular clock in controlling cellular calcium localisation to then impact on calcineurin activity. This in turn affects mitochondrial morphology and metabolism to influence DC antigen processing.

## Circadian variation in mitochondrial calcium is regulated by circadian control of the mitochondrial calcium uniporter

While we demonstrated that calcium localisation displayed a circadian pattern, we wished to understand the mechanism underpinning these effects. As the recently discovered mitochondrial calcium uniporter (MCU) complex is the major calcium transporter of the mitochondria, we reasoned that the molecular clock might regulate components of the MCU complex and control mitochondrial calcium levels throughout the day[43,44]. To investigate this, we isolated CD11c$^+$ cells from mouse spleens at ZT1, ZT7, ZT13 and ZT19. Genes associated with the MCU were analysed by qPCR and their oscillations were investigated for circadian rhythmicity using the JTK_Cycle component within MetaCycle[45]. We confirmed circadian rhythmicity in splenic DCs in both *Bmal1* and *Per2* (Fig. 7a) and also found that genes associated with mitochondrial calcium uptake, *Mcub* and *Emre*, were cycling in a similar circadian pattern and phase. This suggested that circadian control of antigen processing in DCs could be regulated through circadian control of mitochondrial calcium transport. To confirm this, we investigated antigen processing potential of BMDCs in the presence or absence of the MCU inhibitor, ruthenium red[46]. We found a significant decrease in mitochondrial calcium uptake in both *Bmal1$^{+/+}$* and *Bmal1$^{-/-}$* in the presence of ruthenium red (Fig. 7b) and critically, ruthenium red also inhibited the BMDCs potential to process DQ-OVA (Fig. 7c). The inhibition of the mitochondrial calcium uptake with ruthenium red also reduced DC's potential for activation of OTII CD4$^+$ T cells, as indicated by IFNγ levels (Fig. 7d). These data show that components of the MCU complex are regulated in a circadian manner and that this temporal control of calcium transport plays a key mechanistic role in regulating antigen processing within DCs. Accompanying materials and methods are listed in the supplementary section.

## Discussion

The significant finding of this study is that the molecular clock in DCs influences T cell responses. We demonstrate that the DC molecular clock orchestrates a series of events, including altering mitochondrial morphology and coordinating calcium localisation to regulate antigen

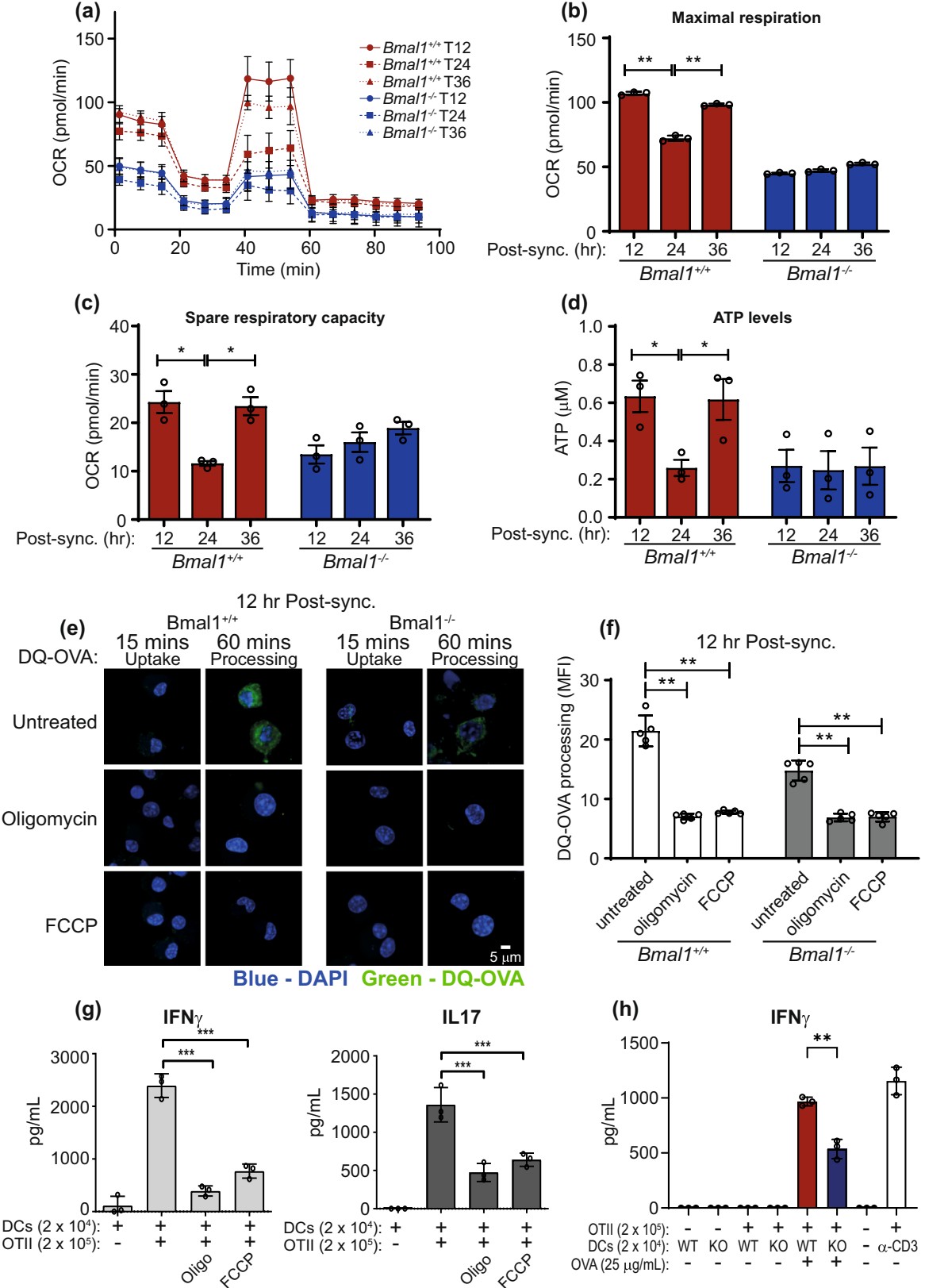

processing and presentation. Any change to cytoplasmic calcium will in turn regulate cytosolic, calcium-dependent enzymes. Calcineurin is one such calcium-dependent phosphatase that is also essential in regulating mitochondrial morphology via DRP1[40]. Furthermore, it has been reported mitochondrial calcium buffering can regulate calcineurin activity[47]. We showed that as mitochondrial calcium increased

during the circadian cycle, cytoplasmic calcium decreased. Reduced cytoplasmic calcium lowered calcineurin's phosphatase activity, thereby promoting mitochondrial fusion. Consistent with our findings increased concentrations of mitochondrial calcium and fusion morphology are both associated with higher mitochondrial metabolism[38,48,49]. Increasing mitochondrial fusion, by inhibition of

**Fig. 3 | Dendritic cell mitochondrial metabolism displays circadian rhythmicity which directs antigen processing and T cell activation. a** *Bmal1*[+/+] and *Bmal1*[−/−] BMDCs were synchronised and OCR was measured at indicated times post serum synchronisation using an XF$_e$96 Analyzer and from this **b** maximal respiration and **c** spare respiratory capacity measurements were derived (*n* = 3 biologically independent samples). **d** *Bmal1*[+/+] and *Bmal1*[−/−] BMDCs were synchronised and ATP levels were measured at indicated times using an ATP/ADP assay kit (*n* = 3 biologically independent cells). **e** and **f** *Bmal1*[+/+] and *Bmal1*[−/−] BMDCs at 12 h post synchronisation were treated with oligomycin (10 μM) or FCCP (10 μM) and antigen processing was then measured by confocal microscopy using DQ-OVA (1 μg/mL) (*n* = 5 biologically independent samples). **g** BMDCs (unsynchronised) were treated with oligomycin (10 μM) and FCCP (10 μM) for 2 h. OVA protein (25 μg/mL) was then added to the BMDCs for 2 h. Supernatants were removed and indicated number of OTII CD4$^+$ T-cells were added. Cells were incubated for 3 days before IFNγ and IL17 were analysed by ELISA (*n* = 3 biologically independent samples) (**h**) *Bmal1*[+/+] and *Bmal1*[−/−] BMDCs (unsynchronised) were incubated with OVA protein (25 μg/mL) for 2 h. Supernatants were removed and indicated number of OTII CD4$^+$ T-cells were added. Cells were incubated for 3 days before IFNγ was analysed by ELISA (*n* = 3 biologically independent samples). Data shown is mean with error bars representing ± SEM. Statistical significance was determined using one-way ANOVA with Tukey's post-hoc test for multiple comparisons. Results are from duplicate BMDCs cultures, from two independent experiments. *$p < 0.05$, **$p < 0.01$, ***$p < 0.001$. Source data are provided as a Source Data file.

---

calcineurin or DRP1, enhanced antigen processing. Therefore, these daily oscillations in mitochondrial morphology and its associated metabolic output impact on the capacity of DC to process and present antigens to T cells.

There is now a growing body of evidence that circadian rhythms play a key role in the immune response to vaccination. It has been shown that immunisation with BCG, influenza or hepatitis A vaccines in the morning resulted in higher antibody responses when compared with vaccination in the afternoon[8,9,27,28]. While it has been shown that T cells proliferated to a greater extent when stimulated either through MHC-TCR or anti-CD3 at ZT8[29,50], these studies used peptides rather than whole protein thereby bypassing antigen processing by DCs[29]. We adoptively transferred CTV$^+$ labelled OTII CD4$^+$ T cells which were harvested from mice at one circadian phase and transferred into mice that were phase shifted to ZT7 or ZT19. Transfer of T cells from one circadian phase into either ZT7 or ZT19 mice allowed us to examine more specifically the DC circadian response in vivo. Our work shows that *Bmal1* control of antigen processing also contributes to the circadian regulation of vaccination. To our knowledge, there are no studies investigating the role of DCs in circadian responses to vaccination. Ex vivo analysis showed a steady increase in DC antigen processing from ZT7 to a peak at ZT19 and a decline. We rationalise that ZT7 injections align with a time that allows optimal antigen processing capacity of DCs. Hence, immunising mice at ZT7 shows a higher response in comparison to ZT19. Our results are consistent with Holtkamp et al. who demonstrate that skin DCs preferentially migrate into lymphatic vessels at ZT7 due to rhythmic gradients in the chemokines CCL21, LYVE, JAM-A and CD99 on skin lymphatic endothelial cells and CCR7, the receptor on DCs for CCL21[22]. Thus, our results, along with others, show that activation of T cell responses is greatest when mice are immunised around ZT7. Taken together, this suggests that the molecular clock of the immune system coordinates DC and T cell functions for optimal activation to invading pathogens at distinct times-of-day.

Significant evidence suggests that circadian rhythms regulate glucose homoeostasis, lipid metabolism and amino acid metabolism at both the cellular and organism level[51–56]. The strong links between circadian control of metabolism provoked our current study given the significant number of studies showing that metabolism is a major regulator of DC function[14,57] in particular via mitochondrial metabolism[24,35]. Similarly to the results on antigen processing, we found significant circadian rhythmicity in DC mitochondrial morphology and metabolism[12,35,57]. Mitochondrial metabolism was essential for antigen processing as electron transport chain inhibitors decreased antigen processing by DCs and their ability to activate of T cells. This places mitochondrial metabolism as an upstream regulator of antigen processing adding to the growing list of immune processes regulated by metabolism.

Altering mitochondrial morphology is one way a cell can alter its mitochondrial metabolism with increased mitochondrial fusion being linked to increased OXPHOS and energy production[38,48]. *Bmal1* dependent rhythms in antigen processing temporally aligned with the rhythms in mitochondrial fusion, implicating a previously uncharacterised role for clock controlled mitochondrial morphology and metabolism in antigen processing. Our data suggests that the molecular clock controls mitochondrial morphology by regulating levels and activity of the key fission proteins, Fis1 and Drp1. In a key rescue experiment we found that the Drp1 inhibitor Mdivi-1 could reverse the circadian decrease in antigen processing back to peak levels[39]. More impressively, increasing mitochondrial fusion in DCs lacking a functional circadian clock also rescued the antigen processing deficit observed in these cells. Increasing antigen processing through promoting mitochondrial fusion demonstrates the mechanistic link between mitochondrial morphology and antigen processing in DCs. Thus, artificially fusing the mitochondria with small molecules could increase antigen processing and might be considered for future vaccine formulations[58].

The ability of circadian rhythms to regulate antigen processing/presenting was intricately linked to calcium localisation. Mitochondria provide calcium buffering capacity to regulate cellular cytosolic calcium[59,60]. Furthermore, mitochondrial Ca$^{2+}$ buffering capacity lowered cytosolic Ca$^{2+}$ that increased in mitochondrial fusion, boosting OXPHOS and ATP production, due to decreased calcineurin activity[40,41,47,61]. Synchronised HepG2 cell cultures display a circadian rhythm in mitochondrial Ca$^{2+}$ which drives rhythmic mitochondrial respiration[17]. Given calcium's importance in energy production, it is perhaps unsurprising it plays an important role in DC antigen processing/presentation. Experiments over the last number of decades with ruthenium red suggested the existence of a mitochondrial calcium transporter, however, the molecular identity of the MCU complex was only recently discovered[43]. Here, we report, that genes associated with the MCU complex are circadian regulated. More specifically, we found that *Emre* and *Mcub*, negative regulators within the MCU complex, were expressed in a circadian manner. In addition to our own findings, we have also identified genes associated with the MCU complex, such as, *Mcu* and *Micu1* in existing circadian datasets[14,62]. As the molecular clock drives an increase in mitochondrial calcium, cytosolic calcium decreases in a circadian pattern and this allows broad circadian regulation of many calcium regulated enzymes while also boosting mitochondrial metabolism. We predict that circadian control of MCU will regulate diverse functions in many cell types. Calcineurin is indirectly circadian regulated through circadian manipulation of calcium location[40]. Inhibition of calcineurin phenocopied the boost of antigen processing seen with Mdivi-1. This was surprising as calcineurin is an inhibitor of T cell activation and proliferation, we show that when inhibited in DCs, it can increase antigen processing and boost T cell activation and proliferation. A number of aspects including exactly how the molecular clock regulates components of the MCU complex are beyond the scope of this study and will require further investigation. Future work should help to determine the mitochondrial metabolites that enhance antigen processing, and whether ATP is a contributing

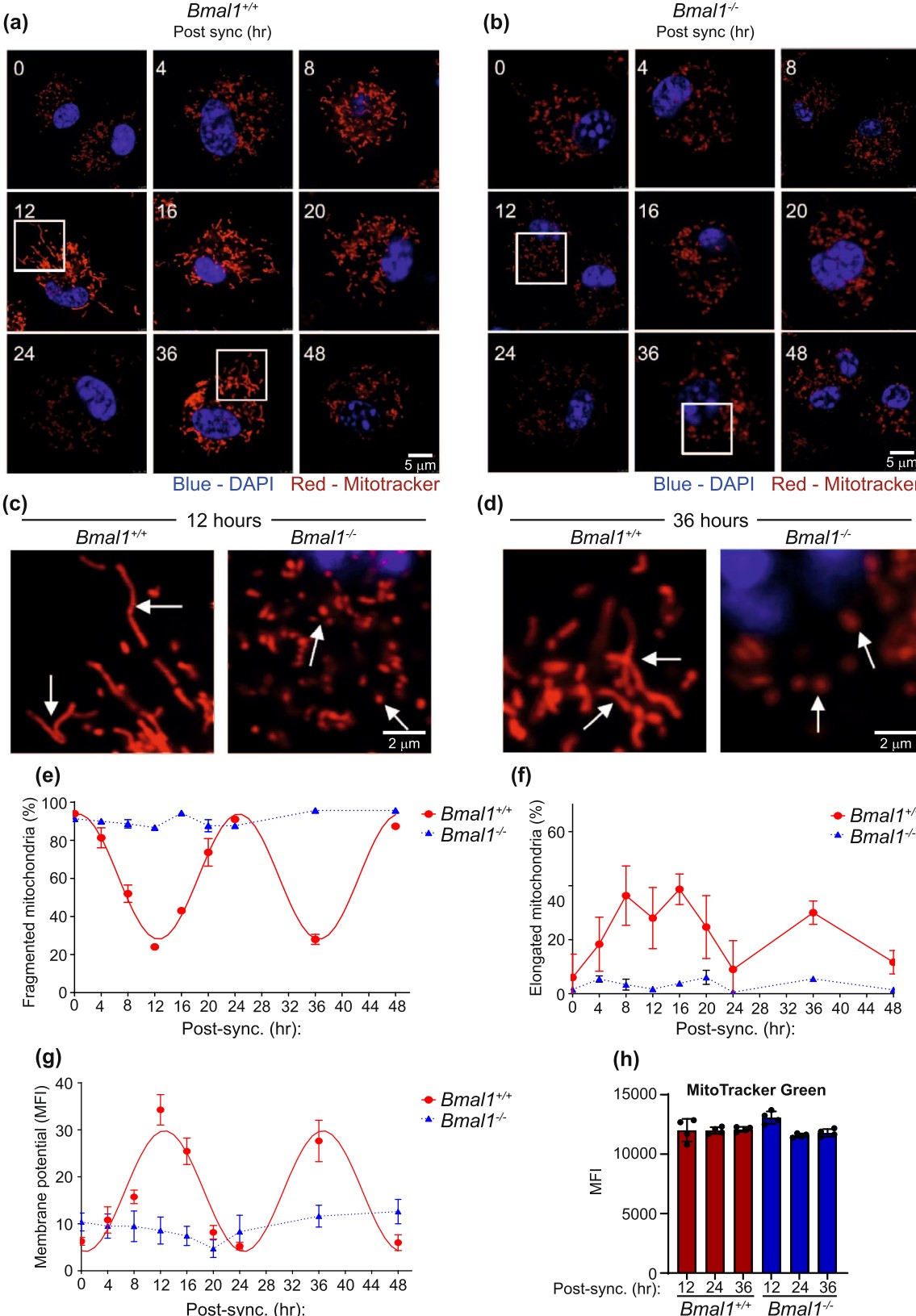

factor. The importance of the ER in terms of mitochondrial $Ca^{2+}$ efflux and antigen processing will also be the subject of future investigations[17].

Collectively, this research has provided insights on the regulation of antigen processing by circadian rhythms and could inform the development of chronotherapeutic vaccination strategies. The finding that mitochondrial morphology and metabolism play a key role in enhancing antigen processing/presentation and T cell activation may enhance vaccination efficacy or longevity. The use of small molecules to modify mitochondrial morphology and metabolism could be a useful approach to enhancing vaccine responses irrespective of time of day.

**Fig. 4 | Mitochondrial fission and fusion in BMDCs display robust circadian rhythms that are dependent on *Bmal1*. a–g** *Bmal1*$^{+/+}$ and *Bmal1*$^{-/-}$ BMDCs were synchronised and mitochondria were stained with Mitotracker Red CMXRos (50 nM) at indicated timepoints. Mitochondrial morphology was assessed over a 48 h time course in **a** *Bmal1*$^{+/+}$ and **b** *Bmal1*$^{-/-}$ using confocal microscopy. Differences in morphology are illustrated in **c** at 12 h and **d** at 36 h post synchronisation between *Bmal1*$^{+/+}$ and *Bmal1*$^{-/-}$ BMDCs. Arrows highlight examples of mitochondrial fusion and fission. (**e**) Mitochondrial fission (fragmented mitochondria) (**f**) mitochondrial fusion (elongated mitochondria and (**g**) mitochondrial membrane

potential were quantified by confocal microscopy ($n = 3$ biologically independent samples). **h** *Bmal1*$^{+/+}$ and *Bmal1*$^{-/-}$ BMDCs were stained with Mitotracker green FM dye at indicated time points post synchronisation and analysed by flow cytometry ($n = 3$ biologically independent samples). Data shown is mean with error bars representing ± SEM. Mitochondrial fission and membrane potential in *Bmal1*$^{+/+}$ were predicted to be circadian by cosinor analysis (**e** and **g**). Data were compared by one-way ANOVA with Tukey's post-hoc test for multiple comparisons (**h**). Source data are provided as a Source Data file.

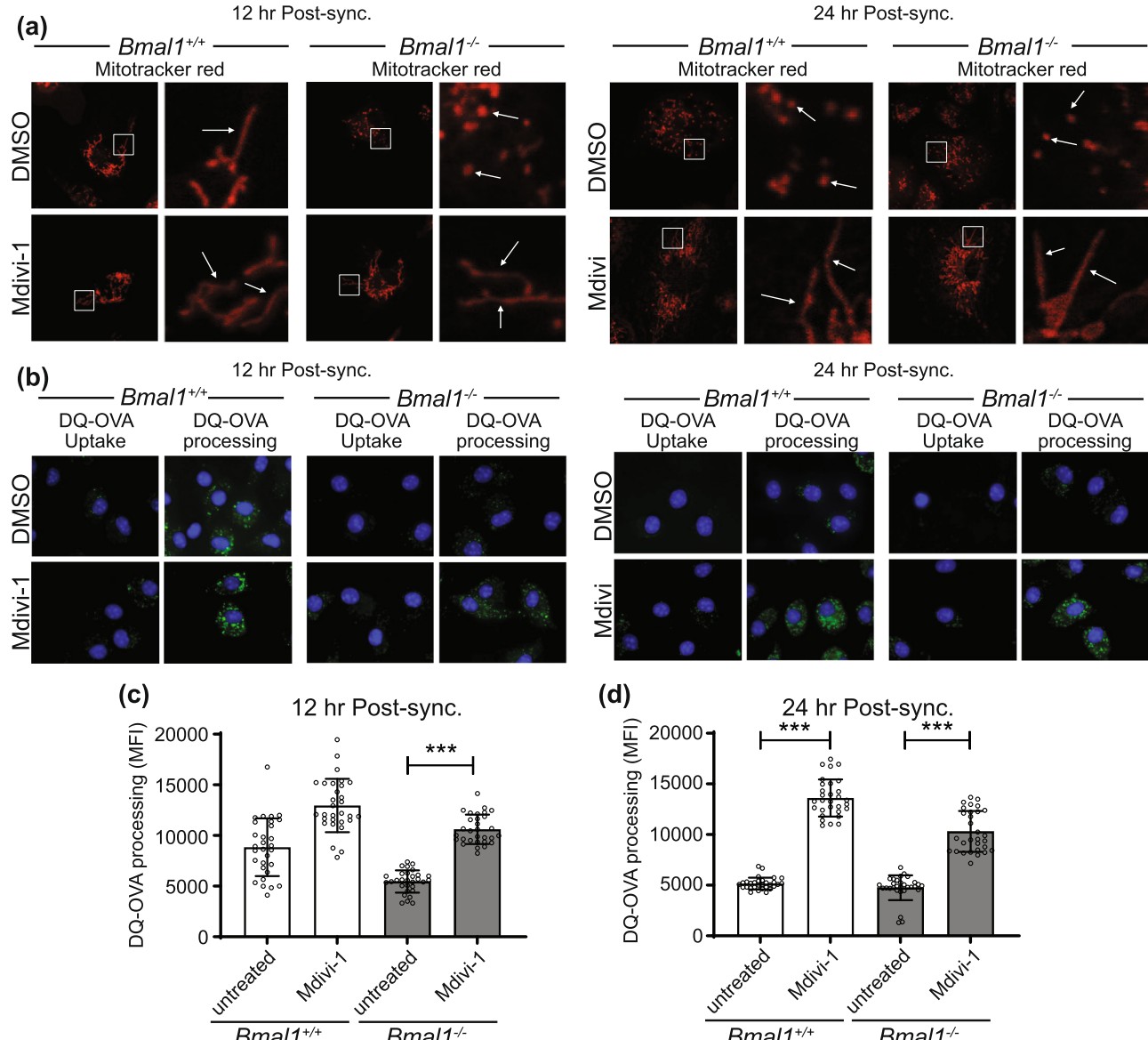

**Fig. 5 | Promoting mitochondrial fusion with Mdivi-1 increases DC antigen processing. a** *Bmal1*$^{+/+}$ and *Bmal1*$^{-/-}$ BMDCs were synchronised and treated with Mdivi-1 (10 μM) 12 h prior to the designated timepoint. At the designated time point, mitochondria were then stained with Mitotracker Red CMXRos (50 nM) and mitochondrial morphology was measured using confocal microscopy. **b** *Bmal1*$^{+/+}$ and *Bmal1*$^{-/-}$ BMDCs were synchronised and treated with Mdivi-1 (10 μM) as in (**a**) and antigen processing was then measured by confocal microscopy at the

indicated times post synchronisation using DQ-OVA (1 μg/mL) (**c** and **d**). Quantification of antigen processing in *Bmal1*$^{+/+}$ and *Bmal1*$^{-/-}$ BMDCs using confocal microscopy at indicated times post-synchronisation ($n = 30$ independent images). Data shown is mean with error bars representing ± SEM. Data were analysed by one-way ANOVA with Tukey's post-hoc test for multiple comparisons. ***$p < 0.001$. Source data are provided as a Source Data file.

## Methods
### Ethical regulations
All research complied with all relevant ethical regulations. All mice were maintained according to European Union regulations and the Irish Health Products Regulatory Authority. Experiments were performed under Health Products Regulatory Authority license with approval from the Trinity College Dublin BioResources Ethics Committee. All animal procedures were in line with the EU Directive 2010/63/EU and with a project authorisation number AE19136/P007.

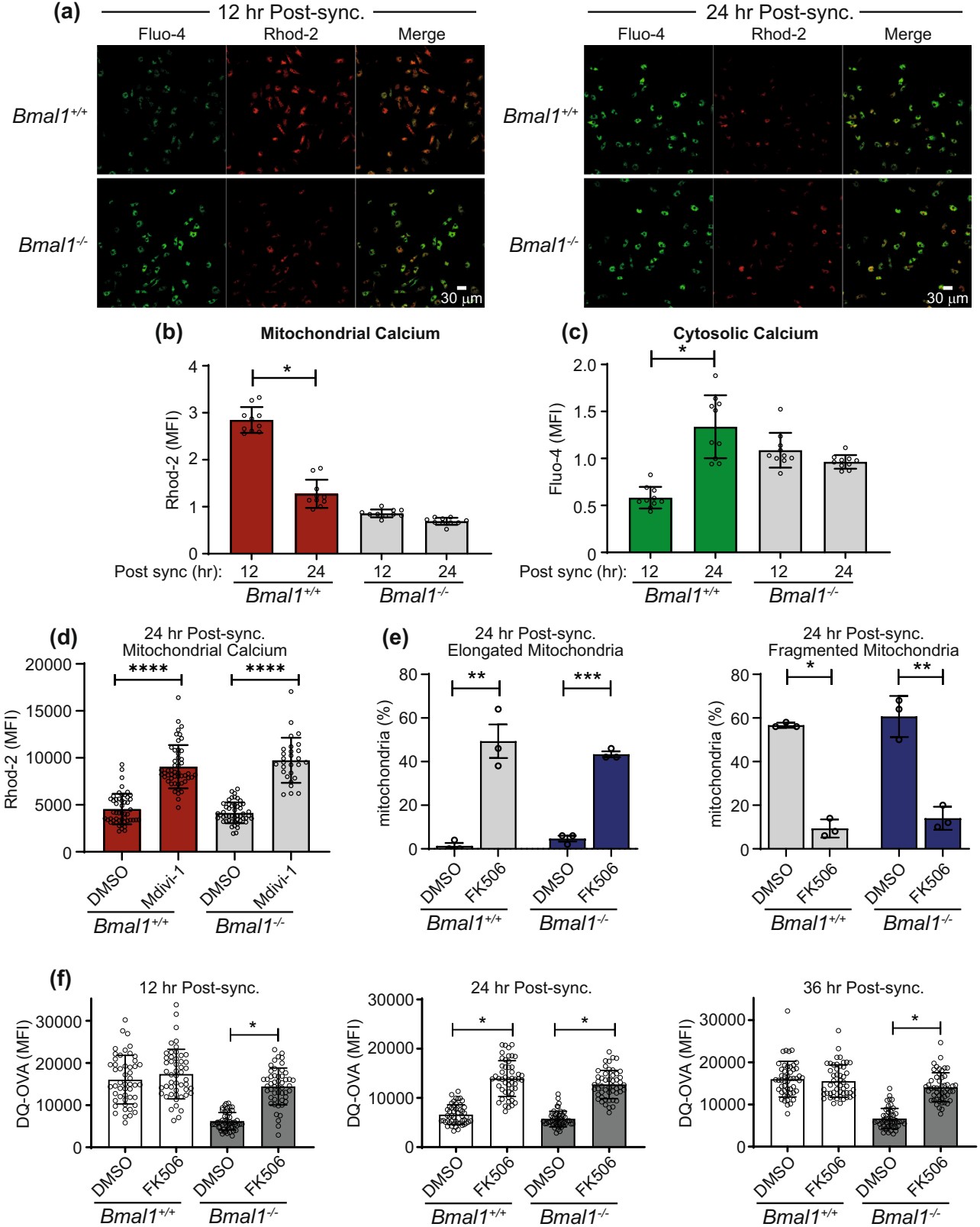

**(a)** 12 hr Post-sync. / 24 hr Post-sync.

**(b)** Mitochondrial Calcium

**(c)** Cytosolic Calcium

**(d)** 24 hr Post-sync. Mitochondrial Calcium

**(e)** 24 hr Post-sync. Elongated Mitochondria / 24 hr Post-sync. Fragmented Mitochondria

**(f)** 12 hr Post-sync. / 24 hr Post-sync. / 36 hr Post-sync.

## Mice

All mice were 6–10 weeks old at the initiation of experiments and housed in a specific pathogen–free facility in the Comparative Medicine Unit (CMU), Trinity College Dublin. The environmental conditions maintained in all rodent rooms in CMU were temperature 20–24 °C with 45–65% humidity. All mice were on a C57BL/6 J background, with wild-type mice obtained from the CMU at Trinity College Dublin. For the adoptive transfer experiments (Fig. 1) only male mice were used. Mice with the gene *Bmal1* containing LoxP sites were kindly provided by Christopher Bradfield. *Bmal1LoxP/LoxP* were crossed with Lyz2Cre mice, which express Cre recombinase under the control of the Lyz2 promoter to produce progeny that have *Bmal1* excised in the myeloid lineage. *Bmal1LoxP/LoxP::Lyz2Cre* (*Bmal1^myeloid−/−^*) mice where compared with control *Lyz2Cre* (*Bmal1^myeloid+/+^*). Offspring were genotyped

**Fig. 6 | Circadian rhythms in calcium localisation co-ordinate rhythms in antigen processing. a** Confocal microscopy analysis of calcium localisation in synchronised $Bmal1^{+/+}$ and $Bmal1^{-/-}$ BMDCs by staining with Fluo-4 (cytosolic) or Rhod-2 (mitochondrial). **b** Mitochondrial calcium quantification and **c** cytosolic calcium quantification from confocal analysis ($n = 10$ independent images). **d** $Bmal1^{+/+}$ and $Bmal1^{-/-}$ BMDCs were synchronised and pre-treated with Mdivi-1 (10 µM). Mitochondrial calcium uptake was quantified using Rhod-2 by confocal microscopy at 24 h post synchronisation ($n = 28$–50 independent images). **e** $Bmal1^{+/+}$ and $Bmal1^{-/-}$ BMDCs were synchronised and treated with FK506 (12 h;

1 µM) and mitochondria morphology quantified by confocal microscopy at 24 h post synchronisation ($n = 3$ biologically independent samples). **f** $Bmal1^{+/+}$ and $Bmal1^{-/-}$ BMDCs were synchronised and antigen processing was quantified by confocal microscopy at indicated timepoints by the addition of DQ-OVA (1 µg/mL) in the presence or absence of FK506 (1 µM). ($n = 50$ independent images). Data shown is mean with error bars representing ± SEM. Data were analysed by one-way ANOVA with Tukey's post-hoc test for multiple comparisons. *$p < 0.05$, **$p < 0.01$ and ****$p < 0.0001$. Source data are provided as a Source Data file.

to confirm the presence of LoxP sites and Cre recombinase. C57Bl/6J mice, OVA-specific CD4+ (OT-II) T-cell receptor-transgenic mice ($H$-$2b$)[63] for the adoptive transfer of OT-II cells and PERIOD2::luciferase[64] were also bred in CMU. All mice were maintained on a 12 h:12 h light:dark regimen with ad libitum food and water prior to experimentation.

**Isolation and staining of OT-II cells for adoptive transfer experiments.** OT-II transgenic mice (B6.Cg-Tg(TcraTcrb)425Cbn/J) were euthanized, and spleens were removed by dissection. Tissues were mashed onto a cell strainer, and the cells obtained were pooled, washed twice in PBS solution, and resuspended in PBS at $1 \times 10^8$ cells/ml. OTII CD4$^+$ T cells were isolated by negative selection, using EasySep magnetic nanoparticles (StemCell Technologies), according to the manufacturer's protocol. The purity of the CD4$^+$ cell population in the enriched fraction was >95%, as determined by flow cytometry analysis. CD4$^+$ isolated T cells were pooled and stained with CellTrace™ Violet (CTV; 5 µM Invitrogen) for 20 min at 37 °C. $4 \times 10^6$ of CTV-labelled T cells were transferred by intraperitoneal (i.p.) injections to mice phase shifted to ZT7 or ZT19 (described in more detail in next section).

**Adoptive transfer of CTV-labelled T cells, immunisations and isolation of mediastinal lymph nodes to conduct T cell characterisation by flow cytometry.** For the adoptive transfer experiment (Fig. 1), light cabinets were used to shift animals to specific light-dark cycles. Mice were given at least two weeks to entrain to any altered lighting schedule changes. This allowed us to transfer CTV-labelled OT-II T cells from donor mice taken at ZT3 into recipient mice who had been phase shifted to ZT7 or ZT19. This allowed simultaneous experimentation at ZT7 and ZT19. 24 h later ZT7/ZT19 mice were immunised simultaneously with 1/50 human dose of a whole cell Pertussis (wcP) vaccine (Shan-5, Shantha Biotechnics Private, India) and OVA (EndoFit Ovalbumin, InvivoGen) 10 µg/mouse. Mice were immunised intraperitoneally (i.p.) with a final volume of 200 µl. 72 h later, again corresponding to the respective phase of ZT7 and ZT19, mice were euthanised by CO$_2$ and mediastinal lymph nodes were harvested.

Mediastinal lymph nodes were passed through a 40 µm and 70 µm cell strainer to a obtain single-cell suspension. Cells were incubated with Zombie NIR™ Fixable Viability kit (Biolegend), for 20 min and then washed with PBS, followed by surface staining with fluorochrome-conjugated anti-mouse antibodies for various markers. Cells were incubated with the antibodies CD69-FITC (H1.2F3), CD11c-BV605 (N418), CD4-BV785 (RM4-5), CD11b-PE-Dazzle™594 (M1/70), CD45R-PE-Cy5 (RA3-6B2) from Biolegend, MHC-II-APC (M5/114.15.2), CD3-PE (145-2C11), CD8-PECy7 (53–6.7) from Thermo Scientific, and with CD16/CD32 FcγRIII (BD Pharmingen) to block IgG Fc receptors. Then cells were fixed in 2% PFA (Thermo Scientific) for 15 min on ice. Flow cytometric analysis was performed on an Cytek® Aurora, and data were acquired using SpectroFlo® software (Cytek Biosciences). The results were analysed using FlowJo software (TreeStar).

**Bone marrow-derived DCs (BMDCs)**
Bone marrow cells were obtained from the femurs and tibiae of 6–10 week old mice of both sexes. This included $Bmal1LoxP/LoxP::Lyz2Cre$ ($Bmal1^{myeloid-/-}$) which where compared with control $Lyz2Cre$

($Bmal1^{myeloid+/+}$). This also included bone marrow harvest from PERIOD2::luciferase mice which were used for lumicycle analysis and also C57Bl/6J mice as WT BMDCs. Bone marrow cells were cultured in Dulbecco's modified eagle medium (DMEM) medium (Gibco) supplemented with 10% foetal bovine serum (FBS) (Gibco), 100 U/ml penicillin, 100 µg/ml streptomycin, and 20 ng/ml GM-CSF (Biolegend, San Diego CA, USA) or 10% J558 cultured supernatants. Cells were maintained at 37 °C in a 5% CO$_2$ atmosphere for 7 days, to allow for cell differentiation into BMDCs. Culture medium was freshly replaced every 2–3 days.

**BMDC synchronisation**
In order to detect circadian rhythmicity at the population levels, cells were synchronised by serum shock treatment. BMDCs ($0.5 \times 10^6$) were cultured in 12-well culture plates overnight in either RPMI or DMEM medium and supplemented with 10% FBS, 1% Penicillin-Streptomycin (100 U/ml), 1% sodium pyruvate and 20 ng/ml GM-CSF.The following day cells were incubated with RPMI medium supplemented with 50% of horse serum (Gibco) for 2 h and then replaced with fresh 5% FBS DMEM, as described previously[30]. BMDCs were considered at 0 h post synchronisation following removal of 50% horse serum. BMDCs from an individual well ($0.5 \times 10^6$ cells) were harvested at indicated time points post synchronisation.

**Lumicycle analysis**
BMDCs from PERIOD2::luciferase mice were plated at a density of $1.8 \times 10^6$ in 35 mm dishes in RPMI medium and supplemented with 10% FBS, 1% Penicillin-Streptomycin (100 U/ml), 10 mM HEPES and 20 ng/ml GM-CSF. The following day, BMDCs were synchronised by serum shock as described above. To monitor circadian rhythmicity, synchronisation media was replaced with Lumicycle recording media (DMEM containing l-glutamine and 1000 mg glucose, without phenol red and sodium bicarbonate (Sigma product code D-2902), 10% FBS, 1% Pen/Strep, 10 mM HEPES, GM-CSF; 20 ng/mL, 0.1 mM beetle luciferin potassium salt (Promega E1603). The 35 mm dishes were sealed using 40 mm coverslips with Dow Corning® high-vacuum silicone grease. Bioluminescence was recorded with the 32-channel Lumicycler by Actimetrics for 5 days beginning at 16 h post serum shock. Analysis was performed using the Actimetrics Lumicycle Analysis programme.

**Harvesting and dissociation of spleen for cell isolation**
$Bmal1^{myeloid-/-}$, $Bmal1^{myeloid+/+}$ and C57Bl/6J mice, were maintained on 12 h:12 h light:dark regimen with ad libitum food and water prior. Spleens were harvested at the designated ZT in RPMI medium with 2 mM L-glutamine (Gibco) supplemented with 10% FBS (Sigma Aldrich) and 100 U/ml penicillin and 100 mg/ml streptomycin (Sigma Aldrich). Spleens were homogenised and passed through a 70 µm cell strainer to generate a single cell suspension and used for subsequent analysis.

**Antigen processing assays**
Antigen processing assays were performed as described using DQ-OVA[65] and were analysed either by confocal microscopy or flow cytometry. DQ-OVA is a self-quenched conjugate of the ovalbumin protein which is strongly labelled with BODIPY dyes, it will exhibit bright

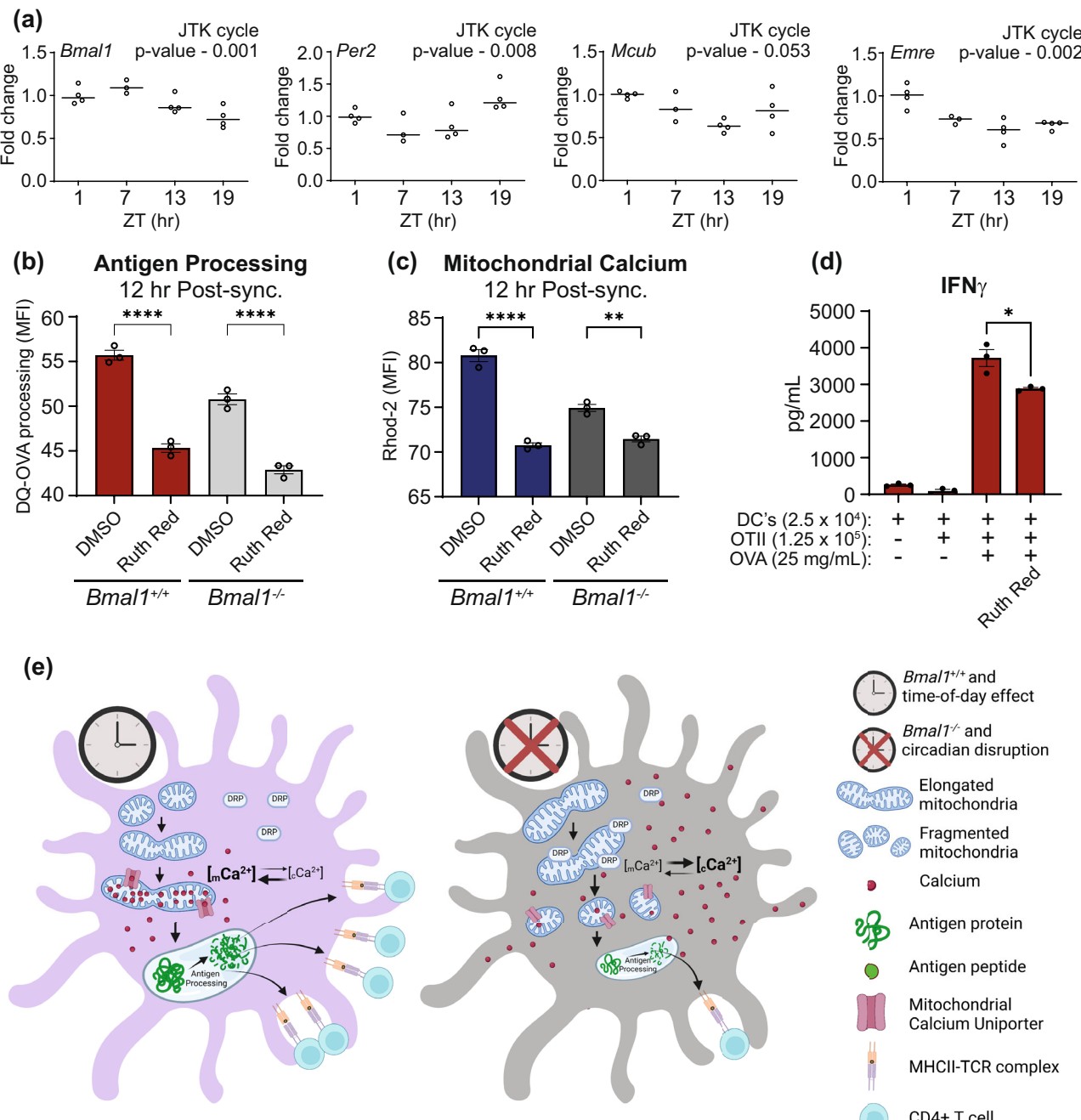

**Fig. 7 | Circadian variation in mitochondrial calcium and antigen processing is directed via control of the mitochondrial calcium uniporter. a** Spleens were isolated from WT mice at ZT 1, 7, 13 and 19. CD11c⁺ cells were isolated and mRNA analysed by qPCR. Circadian analysis was performed using Metacycle and cycMethod set to "JTK". *P* value for each gene is specified on the graph. (*n* = 3 mice) (**b, c**) *Bmal1⁺/⁺* and *Bmal1⁻/⁻* BMDCs were synchronised by serum shock. DQ-OVA and mitochondrial calcium uptake was quantified at 12 h post synchronisation in the presence and absence of ruthenium red (5 μM) (*n* = 3 biologically independent samples). **d** CD11c⁺ cells were isolated from WT spleen at ZT4 and treated with

ruthenium red (10 μM) for 3 h. OVA protein (25 μg/mL) was then added for 2 h. Supernatants were removed and indicated number of OTII CD4⁺ T-cells were added to CD11c⁺ cells. Cells were incubated for 3 days before IFNγ were analysed by ELISA (*n* = 3 biologically independent samples) *p* = 0.02. **e** Schematic showing proposed mechanisms by which the circadian clock in DCs controls antigen processing as inferred from the present study. Data shown are means with error bars representing ± SEM. Data were analysed by Ordinary one-way ANOVA with Tukey's post-hoc test for multiple comparisons (**b, c**) or by a two-tailed *t*-test (**d**). **\**p* < 0.01 and **\*\*\*\**p* < 0.0001. Source data are provided as a Source Data file.

fluorescence upon proteolytic degradation into singly, dye labelled peptides. Antigen processing by DQ-OVA was analysed either by confocal microscopy or flow cytometry.

For confocal microscopy analysis, BMDCs (2 × 10⁵ cells/well) were grown in Lab-Tek chambers (Thermo Fisher Scientific) and DQ-OVA (1 μg/mL; Life technologies) was added to cells for 15 min at 37 °C (to measure uptake but not processing and is used as a control) or 60 min

at 37 °C (to measure processing) at indicated time points. At the end of each incubation period, BMDCs were washed and fixed with 4% paraformaldehyde. Fixed cells were mounted with DAPI-containing Vectashield (Vector) and analysed by using a Leica SP8 scanning confocal microscope (Wetzlar, Germany), using a 63× oil immersion objective. Mean Fluorescence Intensity (MFI) of DQ-OVA was assessed on 25–30 images for each experimental condition and from three independent

experiments using ImageJ software (National Institutes of Health, Bethesda, MD).

To investigate the impact of mitochondrial metabolism/function on antigen processing, BMDCs were pre-treated with oligomycin (10 μM) and FCCP (100 μM) for 1 h prior to performing the antigen processing assay and quantification by fluorescence on confocal microscopy.

For flow cytometry analysis, antigen processing analysis of splenic DCs was performed the same way as described above except cells were plated in 96 well plates ($0.5 \times 10^6$) per well. After either 15 min or 60 min of DQ-OVA, cells were fixed with paraformaldehyde and analysed by flow cytometry.

Synchronised BMDCs were pretreated with Mdivi-1 (10 μM–M0119, Sigma) or FK506 (1 μM – TLRL-FK5, Invivogen) for 12 h prior to harvesting.

### B16-FLT3L cell culture and in vivo expansion of splenic DCs

B16-FLT3L cell line was cultured in a humidified 5% $CO_2$ atmosphere at 37 °C with DMEM medium with 2 mM L-glutamine (Invitrogen Biosciences) supplemented with 10% (v/v) heat-inactivated FCS (Labtech, International) and 100 U/ml penicillin and 100 mg/ml streptomycin (Invitrogen/Biosciences). This cell line was tested for absence of mycoplasma using the PCR Mycoplasma Test Kit (AppliChem) according to manufacturer's instructions. For splenic DC expansion, $2.5 \times 10^6$ B16-FLT3L cells in 100 μl of PBS were injected subcutaneously (sc) in the right flank of the mice. Mice were sacrificed 10–13 days after cell injection at the corresponding ZTs and splenic DC subsets and antigen processing analysed by flow cytometry.

### Flow cytometry for ex vivo analysis of splenic DCs

Splenocytes were cultured in U-bottom 96-well plates (VWR) at $1 \times 10^6$ cells/200 μl of supplemented RPMI medium and incubated as described for antigen processing assays. Thereafter, cells were washed and incubated for 5 min at 4 °C with Fc blocking antibody (2.4G2) (BD Pharmingen) before stained for 20 min in the dark at 4 °C with saturating concentrations of surface-targeted antibodies and viability markers. Cells were fixed with Cytofix/Cytoperm kit (BD Biosciences), measured by FACS Fortessa (BD Biosciences) and analysed by FlowJo software (Tree Star, Ashland, OR). Dead cells were excluded using Zombie Yellow (Biolegend). Antibodies used as follows: CD3-APC (145-2c11, Biolegend), F4/80-AF700 (cat. MCA497A700, BioRad), LY6G/6C-APC-Cy7 (RB6-8C5, BD), NK1.1-BV421 (PK136, Biolegend), MHCII-BV711 (M5/114, BD), CD11c-BV785 (N418, Biolegend), CD11b-PE-Cy7 (M1/70, BD), CD45R/B220-V500 (RA3-6B2, BD), CD103-PE (2E7, Invitrogen), CD8-Percp-Cy5.5 (53–6.7, Biolegend) and CD317-BV650 (927, Biolegend).

### Bioenergetic assays and analysis

Oxygen consumption rate (OCR) or mitochondrial respiration was analysed using the Agilent Seahorse XF Cell Mito Stress Test and measured on a $XF_e96$ Analyzer. Briefly, $8 \times 10^4$ BMDCs were seeded in each well, excluding background wells, of an $XF_e96$ cell culture plate and subjected to serum synchronisation for the time points indicated. A utility plate containing the injector ports and probes was filled with calibrant solution and placed in a $CO_2$-free incubator at 37 °C overnight. Before metabolism was measured, the culture medium was removed from cells and replaced with XF assay media pH 7.4 (Agilent). The XF assay media was supplemented with 10 mM glucose, 1 mM pyruvate and 2 mM glutamine. The cell culture plate was then incubated in a $CO_2$-free incubator at 37 °C for 45 min. Oligomycin (1 μM), an ATP synthase (complex V) inhibitor, Carbonyl cyanide-p-trifluoromethoxyphenylhydrazone (FCCP) (0.9 μM), an uncoupling reagent that collapses the proton gradient and Rotenone/Antimycin A (0.5 μM) were added to the appropriate ports of the utility plate for a standard MitoStress test according to the instruction manual (Agilent, 103015-100). This plate was run first on the flux analyser for calibration. Once complete the utility plate was replaced with the cell culture plate and run on the real-time Seahorse $XF_e96$ analyser using the software's Mito Stress template programme.

The Mito Stress test profile allows calculation of the following parameters.

**Basal respiration**: shows energetic demand of the cell under baseline conditions,

**Maximal respiration**: the maximal oxygen consumption rate attained by adding the uncoupler FCCP, which stimulates the respiratory chain to operate at maximum capacity. This measurement shows the maximum rate of respiration that the cell can achieve.

**Spare respiratory capacity**: this measurement indicates the cells ability to respond to an energetic demand, this can be an indicator of cell fitness or flexibility.

BMDCs ($1 \times 10^6$) were plated and synchronised. Cells were treated with FCCP (10 μM) or Oligomycin (10 μM) for 3 h and then harvested at indicated time points post-synchronisation and ATP measured using Abcam ATP assay kit (ab83355).

### Mito Tracker Red CMXRos staining and mitochondrial dynamic analysis

BMDCs ($0.5 \times 10^6$) were plated on a 35 mm, high glass bottom μ-Dish (Ibidi, Germany) and maintained overnight at 37 °C in a 5% $CO_2$ atmosphere. Cells were synchronised by serum shock as previously described. Cells were stained for 30 min with MitoTracker Red CMXRos (50 nM; Life technologies). Cells were washed with PBS followed by addition of fresh medium and cells were imaged on a Leica SP8 scanning confocal microscope (Wetzlar, Germany), with a 63× immersion objective. Cell images were obtained at indicated hours post synchronisation. Automated image analysis was performed in Fiji using custom-written macros. Fiji macros and a short user guide are available as part of the supplementary material online accompanying this article (Supplementary Software 1). In short, confocal slices of MitoTracker-stained macrophages were normalised to the full 16-bit range. Binary masks of approximate cell outlines were generated by intensity thresholding, binary operations to smoothen outlines and fill holes, watershedding to separate touching cells, and manual error correction. Binary masks of mitochondria were determined by band-pass filtering of raw mitochondrial images followed by auto-thresholding, removal of tiny objects, and splitting of touching mitochondria by marker controlled watershed segmentation using MorphoLibJ[66] restricted to maxima candidates predetermined by the detection of prominent maxima in a distance transform image of the masks. For determining morphological characteristics of all mitochondria and of the mitochondria within each single cell, a batch process was performed on a set of experimental images with its masks. The "Analyze Particles" function was used to measure size and shape descriptors of all mitochondria per cell, from which median area, aspect ratio, length, circularity, and total area were determined. Results were saved as spreadsheet files and as ROI overlays to the original images that enabled the look-up between the statistics and images of every single mitochondrion. Mitochondria were then divided into three different categories, based on length, namely as mitochondria of less than 1 μm, 1–3 μm, and greater than 3 μm[67].

### Mitochondrial membrane potential (ΔΨm)

Labelling of mitochondria with MitoTracker Red CMXRos Life technologies is dependent on mitochondrial membrane potential as indicated by the manufacturer´s instructions. Confocal microscopy was used to take images of 25 individual cells from BMDCs at each time point (hr) post-synchronisation. Mean Fluorescence Intensity (MFI) of mitochondrial staining intensity was measured by Image J.

### MitoTracker Green FM staining

Cells were washed with PBS, scraped, transferred to FACS tubes, and centrifuged at $300 \times g$ for 5 min. MitoTracker Green (Invitrogen, cat. no. M7514) was used at a final concentration of 100 nM diluted in PBS containing 1 mM EDTA and 2% FCS (FACS buffer). Fifty microliters of 100 nM MitoTracker Green was added to each sample and incubated for 15 min at 37 degrees Celsius. Cells were washed with 1 mL FACS buffer, centrifuged at $300 \times g$ for 5 min, and resuspended in 200 μL FACS buffer. Data was acquired using FACS Canto II and mean fluorescent intensity (MFI) was obtained through analysis with FlowJo Software (V10.8.1).

### Immunoblot analysis

Cells were lysed in SDS PAGE sample buffer, samples boiled for 7 min, then cooled and loaded on to a SDS polyacrylamide gel for separation by electrophoresis. Following separation, samples were transferred onto nitrocellulose membranes. Membranes were probed with primary antibodies for BMAL1 (cat. #14020S, Cell Signalling Technology, dilution 1:1000), OPA1 (cat. #80471, Cell Signalling Technology, dilution 1:1000), FIS1 (cat. #PA5-22142 Thermo fisher Scientific, dilution 1:1000), MFN1 (cat. # ab126575, Abcam, dilution 1:500), MFN2 (cat. #9482S, Cell Signalling Technology, dilution 1:1000), DRP1 (cat. # 5391S, Cell Signalling Technology, dilution 1:1000), p-DRP1 (S637) (cat. # 4867S. Cell Signalling Technology, dilution 1:1000), α-Tubulin (cat. # 3873S, Cell Signalling Technology, dilution 1:1000) and β-Actin (cat. # MAB1501, EMD Millipore, dilution 1:10,000) followed by incubation with appropriate Peroxidase-conjugated AffiniPure Goat anti-rabbit IgG (cat. # 111-0350144, Jackson ImmunoResearch, dilution 1:2000) or Peroxidase-conjugated AffiniPure Goat anti-mouse IgG (cat. # 115-035-146, Jackson ImmunoResearch, dilution 1:2000). Bands were detected by chemiluminescence using Immobilion Western Chemiluminescent HRP Surbstrate (PVBKLSO500, Sigma). Bands were visualised and quantified using an Amersham 680 Imager (GE Healthcare) and ImageStudioLite and normalised to the intensity of the α-Tubulin or β-Actin band.

### Real-time polymerase chain reaction

Total RNA was isolated using the Invitrogen PureLink RNA Mini Kit (Thermo Fisher, 12183025) and quantified using a Nano-Drop 1000 Spectrophotometer (Thermo Scientific Fisher). cDNA was prepared using 50–100 ng/μl total RNA using a High Capacity cDNA Reverse Transcription Kit (Thermo Fisher, 4368813), according to the manufacturer's instructions. Primers were designed using the NCBI database (https://ncbi.nlm.nih.gov/tools/primer-blast) and provided by Eurofins. Please refer to Supplementary Table 1 for a list of primers used. Real-time quantitative PCR (RT-PCR) was performed on cDNA, diluted 1 in 2 with RNAase-free water, using SYBR Green probes on a 7900 HT Real-Time PCR System (Applied Biosystems). Fold changes in expression were calculated by the Delta-Delta ($\Delta\Delta$) Ct method using 18 s as a control for mRNA expression. All fold changes were normalised to untreated/non-targeting controls.

### Calcium localisation analysis

BMDCs were plated in glass bottom plates (Ibidi) and synchronised, as previously described. At 12 h and 24 h post synchronisation, BMDCs were washed with PBS and labelled with either Fluo-4 AM (8 μM) (Invitrogen) rhodamine-2 AM (4 μM) (Invitrogen), which are cytosolic and mitochondrial calcium indicators respectively, for 1 h at 37 °C in 5% $CO_2$. Following incubation with calcium stains, cells were rinsed six to seven times with cold PBS, replenished with fresh DMEM and incubated for 20 min before imaging. $Ca^{2+}$ imaging was conducted at room temperature on a Leica SP8 scanning confocal microscope (Wetzlar, Germany), using a 20× objective. Cell images were analysed with the ImageJ software.

### DC and CD4 T cell coculture

BMDCs or splenic DCs (CD11c[+]) were plated at $2.5 \times 10^4$ cells per well in a U-bottomed 96 well plate. BMDCs were treated with oligomycin (10 μM), Trifluoromethyoxy carbonlcyanide phenylhydraxone (FCCP; 10 μM) or ruthenium red (10 μM) for 3 h before addition of Ovalbumin (25 μg/mL) for 2 h followed by treatment with LPS (10 ng/mL) for 4 h (or remainder of experiment). Cells were then washed with PBS to remove inhibitors from culture medium and prevent any carry over effects on T cells. CD4+ T cells were isolated from spleens of OTII transgenic mice using CD4 negative selection kit (Stem Cell) according to manufacturer's instructions. $1.25 \times 10^5$ CD4[+] T cells were then added to the wells containing treated BMDCs or splenic DCs (CD11c+) and incubated for 3 days. Supernatants were collected on day 3 and analysed for IFNγ and IL17 by ELISA (RnD systems).

### Circadian data analysis

Mitochondrial morphology, membrane potential, antigen processing in cultured BMDCs were investigated for the presence of circadian patterns using multiple regression to fit a linearised cosinor model with a pre-determined period of 24 h. Circadian patterns were indicated by statistical significance of the predicted cosinor (sine and cosine) regression coefficients[68]. The cosinor model was defined by linear sine and cosine terms of transformations of the time variable in hours:

$$Y_i = M + \beta \, Cos\,(2\pi t_i/24) + \gamma \, Sin\,(2\pi t_i/24) \tag{1}$$

Where Y is time in hours post serum shock, and $M$, $\beta$ and $\gamma$ were predicted by regression (1). The intercept (M) is the mean level of the curve predicted from Eq. (1), and the acrophase ($\Phi$) is the peak x axis value of the curve, calculated as:

$$\Phi = \tan^{-1}(-\gamma/\beta) \tag{2}$$

The amplitude (A) was the distance from the mean to the acrophase, providing an estimate of the magnitude of rhythmicity.

$$A = (\beta^2 + \gamma^2)^{1/2} \tag{3}$$

The cosinor curve provided a graphical representation of how closely the data approximated to the 24 h periodicity of a circadian dataset, and the statistical significance of this was determined by testing the null hypothesis that the amplitude of the curve was equal to zero. Statistical significance was accepted at values of $p < 0.05$ and all analyses were performed using Stata 14 statistical software (StataCorp, College Station, TX).

MetaCycle was used to detect the presence of circadian rhythms in lumicycle data from synchronised PER2::Luciferase BMDCs and mRNA expression in splenocytes over the 24 h cycle. In MetaCycle, default settings and period length were set to 24 (h) for min and max period, cycMethod was set to "JTK" and circadian rhythms were identified with statistical significance $p < 0.05$.

### Statistical and reproducibility

GraphPad Prism 8.00 (GraphPad Software) was used for statistical analysis. A one-way ANOVA test was used for the comparison of more than two groups, with Tukey test for multiple comparisons. A two-tailed Student's $t$-test was used when there were only two groups for analysis. All error bars represent SEM. Significance was defined as $*p < 0.05$, $**p < 0.01$, $***p < 0.001$, $****p < 0.0001$. Any specific statistical tests and details of 'n' numbers done for experiments are listed under the corresponding figures. Measurements were taken from distinct

samples. No statistical method was used to predetermine sample size. No data were excluded from the analysis. The experiments were not randomised and Investigators were not blinded to allocation during experiments and outcome assessment.

## Reporting summary

Further information on research design is available in the Nature Portfolio Reporting Summary linked to this article.

## Data availability

Source data are provided with this paper.

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

## Acknowledgements

The majority of this study was supported through the funding provided by Science Foundation Ireland through a Career Development Award (17/CDA/4688) and by the Irish Research Council through a Laureate Award (IRCLA/2017/110) and an RCSI Strategic Academic Recruitment Program (StAR) award all provided to AMC. Further support was provided by a Conacyt grant to MCS (CVU440823), a Science Foundation Ireland Investigator Award (16/IA/4468) to K.H.G.M. and a European Research Council Consolidator Award (ERC-CoG_771419) to D.K.F. Flow cytometry was performed at the Science Foundation Ireland funded Flow Cytometry Facility at Trinity Biomedical Sciences Institute. We wish to acknowledge the laboratory operations staff within the RCSI School of Pharmacy and Biomolecular Sciences for their technical assistance throughout the project. Figures 1a, 2a and 7e were created with BioRender.com.

## Author contributions

M.P.C.S. and R.G.C. conceived, designed and performed the majority of experiments, analysed the data and wrote and edited the manuscript. M.M.W., D.M., D.K.F. and K.H.G.M. assisted in the design, execution and analysis of the in vivo experiments. C.A.P. performed experiments with P110 and Mdivi-1, J.R.O.S. performed and analysed luciferase reporter experiments. J.R.O.S., S.M.B. and J.M.H. provided bioinformatic expertise and analysis. S.L.C. conducted mitochondrial mass experiments. T.D., L.E.F., P.A.K., R.J.S.P., G.A.T. and J.O.E. assisted in qPCR, western blot, bioenergetic, antigen processing and flow cytometry experiments and analysis. I.S. developed the semi-automated programme for mitochondria analysis. Y.H. and A.M. provided *Bmal1*^myeloid–/– mice and BMDC generation. J.S.G. conceived the mitochondrial morphology and antigen processing studies. K.H.G.M. assisted in editing the manuscript. A.M.C. led the project, acquired the funding for the project, conceived and designed experiments, wrote and edited the manuscript.

## Competing interests

The authors declare no competing interests.
