## [Peer Review File · Nature Communications]

The circadian clock influences T cell responses to vaccination
by regulating dendritic cell antigen processingREVIEWER COMMENTS

Reviewer #1 (Remarks to the Author):

The manuscript “The circadian clock influences T cell responses to vaccination by regulating dendritic cell antigen processing through alterations in mitochondrial morphology and metabolism”, by Curtis et al., describes a mechanism by which calcium flux inside the cell (mitochondria/cytosol) can modulate the processing and presentation of antigens from DC to T cells in a circadian way, which could be the responsible for differential responses to vaccination. The study is interesting, noteworthy and well conducted. There are, however, a few concerns that demand specific explanations that could improve the manuscript.

Specific remarks:

- 1) Line 107-108: the sentence “As the transferred T cells were isolated from mice at the same circadian phase and the recipient mice were phase shifted as they were housed under shifted light schedules” is unclear. When the authors mention the isolation of transferred T cells, do they refer to their obtention from the original mice? Or is it the isolation from the experimental mice in order to analyze their proliferation?
- 2) On the other hand, the information of the specific lighting schedules, should be provided in the method section. It is unclear which mice are under the shifted lights schedule (those were injected at ZT7, or at ZT19, or both). It is also unclear which “circadian phase” is being mentioned.
- 3) The legend to Figure 1 states that “Proliferation of CTV stained OTII CD4+ T cells harvested on Day 5 from mediastinal lymph node at ZT7 and ZT19”. In addition to immunization of mice at different times, were the tissues extracted at different times? This data is omitted in the experimental design. Indeed, in order to clarify the circadian organization of the experimental approach, a specific section should be added to the methods section.
- 4) Lines 114-116: as shown in the Figure Suppl 1a, the analysis of CD69 was performed on the CTV-negative cells. The rationale for this should be better explained.
- 5) Suppl. Fig. 1a-b: in the gating strategy used for lymph nodes, the presence of many CD8+CD4+ cells is striking since, theoretically, these cells mainly differentiate in the thymus. Additionally, the fluorescence signal of the CD8 marker is quite high, thus, for a better visualization of the cells that are positive for each marker, it is necessary to show the corresponding isotype controls. On the other hand, this figure should be also cited in the corresponding method sections.
- 6) Since some mechanisms are induced at ZT7 (Fig. 1d) and others at ZT19 (Fig. 2f, g), it is relevant to discuss this apparent contradiction. This appears to be relevant because the antigen processing is necessary for CD4 cells activation.
- 7) The expression pattern of Nr1d1 is different in Suppl. Fig. 2 and 3. Suppl. Fig. 2 shows maximum levels 16-20 h post synchronization and minimal levels at 24 h (very similar to levels at 12h) post synchronization, but Suppl- Fig. 3 shows higher levels at 24 h than at 12 h post synchronization. Authors should revise these data.
- 8) Lines 268-270: The result obtained with the mitochondrial fission inhibitor P110 (Sup. fig. 4a and b) should be further discussed since the data is somewhat contradictory.
- 9) Line 295-296: It is confusing that the circadian analyses of mRNA levels shown in Fig. 7 was performed in a different way than the patterns shown in the remaining figures (particularly Suppl. Fig 2). Also, the explanation of the analyses performed with these data and the meaning of each color line in the figure is missing. In addition, 4 time points are not completely enough to calculate a circadian period. Finally, it is difficult to understand why its

values are different from 24 hours since mice are housed under a standard light-dark schedule.

10) I suggest including a final graphical figure that includes obtained findings to better understand the mechanism of circadian regulation of antigen processing.

Minor concerns

1) Please refer to the methods (supplementary materials) in the main body of the manuscript.

2) Fig 1: it is unclear which statistical test that was used to analyze this data: when did the authors apply paired T-test and when Two-way ANOVA?

3) Fig 2: Is it unclear why pDC and macrophages are shown in the Suppl. Fig. 2, in particular because the response of total CD11c (and CD subtypes) and CD11b are shown in Fig. 2. I suggest including these graphs in Fig. 2.

4) There are figures (like Fig 2c, 3e-h, 6d-e, 7b-d) that omit the time data. It is relevant to mention if this comparison compiled all time points or whether a single time point is shown.

5) Line 176 and Fig. 3: It is necessary to mention how the spare respiratory capacity is calculated. It could be explained in the legend of the figure or in the methods section and mentioned in the result section.

Fig. 3e, f: the description of the result obtained with *Bmal1*^{-/-} cells is missing.

6) Supplementary Materials:

a- Methods:

- "Flow cytometry for ex vivo analysis of splenic cells", line 1: the paragraph also refers to lymph nodes.

- "Data analyses", line 8: it is confusing to refer to dexamethasone treatment for the synchronization step since all methods use serum shock for it. In the same line, the symbols (beta and gamma) are wrong.

- The explanation of migratory and resident DC identification is missing.

- Suppl. Fig 2: the terms Nr1d1 and Rev-erba are used for the same molecule, it is better to call it with the same nomenclature.

b- Figs. 2 and 6 and Suppl. Figs. 2 and 3: the title of the figures is missing.

Reviewer #2 (Remarks to the Author):

In the study titled "The circadian clock influences T cell responses to vaccination by regulating dendritic cell antigen processing through alterations in mitochondrial morphology and metabolism" Annie M. Curtis and co-authors investigate the role of the biological clock and mainly of *Bmal1* in determining T cell responses in a mouse model. T cell responses were greater in the case mice were immunised in the middle of their rest phase. Rhythms in both mitochondrial morphology and metabolism, driven by *Bmal1*, associated with 24-hour rhythms in antigen processing and mitochondrial Ca^{2+} circadian fluctuations. Boosting mitochondrial fusion (inhibiting fission) with Mdivi-1 recovered the circadian deficit in antigen

processing upon Bmal1 KO. They conclude that rhythms in antigen processing are related to rhythmic changes in mitochondrial calcium and subsequently in mitochondrial morphology.

The authors investigate the interplay between circadian rhythms and adaptive immunity focusing on the role of dendritic cells (DC) in this context. In the presented manuscript the authors show evidence of a causal linkage between the Bmal1-mediated rhythmic changes of the mitochondrial bioenergetic activity and the antigen processing in DC. This is not surprising given the already known control exerted by the clock gene machinery on cellular metabolism. In the case of the antigen presentation, the protein antigen is degraded in peptides by the proteasome that, indeed, requires ATP for proper functioning. Of interest is the mechanism proposed by the author whereby the rhythmic activity of the mitochondrial respiration appears linked to changes in the organelle fusion/fission dynamics with fused elongated mitochondria and fragmented mitochondria varying in phase with the high and low respiratory activities respectively. Upstream of this would be the redistribution of calcium ions between the cytosol and the mitochondria that also follows circadian rhythmicity consistent with the alternating expression of the mitochondrial calcium uniporter (MCU). Calcium ions are known to activate the phosphatase calcineurin that in turn modulate the activity of factors involved in the mitochondrial fission machinery. The manuscript is well written, and the results obtained clearly commented and of interest however the following points are raised to which the authors are invited to reply.

The paper only partly contains new information. A recent article addressed the role of the biological clock in the dynamics of dendritic cells, even if in a different context, i.e. migration from the skin to lymph nodes, and should be cited and discussed in the light of the reported results (Holtkamp SJ, ...Scheiermann, C (2021). Circadian clocks guide dendritic cells into skin lymphatics. *Nature immunology*, 22(11), 1375–1381. <https://doi.org/10.1038/s41590-021-01040-x>)

The study design is not totally clear and sound and not all potential biases have been appropriately addressed. The authors mainly centre their study design on only a peculiar immunization methodology and a single not well described cellular model. Besides, the methods are not ever appropriate, not well described and not sufficient details are provided.

Mice were maintained in a 12h:12h light:dark environment. To explore the genuine circadian nature of processes, mice should be housed in constant darkness to avoid masking effects. Please, explain the reasons that lead you to a different choice of housing schedule

Please, describe in more detail how DCs were chosen/isolated/sorted, which kind of OVA peptide was used and how they were pre-loaded with OVA peptide in vitro. Why they did not use the OVA octameric peptide?

“As the transferred T cells were isolated from mice at the same circadian phase and the recipient mice were phase shifted as they were housed under shifted light schedules. We reasoned that this approach allowed us to specifically measure the effect of the molecular clock in DCs on T cell activation” Please, clarify/correct/discuss in depth this statement, critical for your study design appropriateness

The authors should perform ex vivo re-challenge with OVA of mediastinal lymph node derived T cells to analyze cytokines secretion

How PER2::Luc bone marrow derived dendritic cells (BMDCs) were produced ?

Analysis of expression of the molecular clock genes, Period 2, Nr1d1 and Bmal1 in BMDCs by qPCR should be confirmed with western blotting of the same proteins

The authors state “Bmal1 is the only non-redundant clock gene, and its deletion ablates all molecular clock function”. This is not correct...refer to <https://doi.org/10.1016/j.cub.2009.12.034> and <https://doi.org/10.1126/science.aaw7365>

A better characterization at transcriptomic level should be performed, with scRNA-seq on Bmal1^{+/+} and Bmal1^{-/-} CD11b⁺ cells, without and with OVA stimulation, with cell synchronization and followed by comprehensive bioinformatics analysis.

Considering that chromatin regulators controlling rhythmicity manage the interplay between the biological clock and the immune system, epigenome remodeling evaluation with differential chromatin accessibility analysis should also be performed on these cell types (Bmal1^{+/+} and Bmal1^{-/-} CD11b⁺ cells)

There is an error in the labels in figure 2E, CD11b⁺ is repeated two times

The authors use Mdivi-1, a quinazolinone derivative, as an inhibitor of dynamin-related protein 1 (DRP1), a crucial driver of mitochondrial fission, to inhibit Drp1-dependent mitochondrial fission and indirectly boost fusion-related patterns of mitochondrial respiration. Anyway, the results of this experimental protocol are absolutely expected, not particularly informative.

The authors should take in account that in vivo DRP1 is phosphorylated with circadian rhythmicity influencing mitochondrial morphology and the biological clock mechanism. This series of effects could greatly lessen valuability of the results if DRP-1 inhibition is not performed at different time intervals. More informative could be the use of phosphates in a P-Drp1 rescue experiment

Again, a better characterization at transcriptomic level should be performed, with scRNA-seq on Bmal1^{+/+} and Bmal1^{-/-} BMDCs without and with OVA stimulation, with cell synchronization and followed by comprehensive bioinformatics analysis.

The authors report the following statement “We also found that antigen processing was reduced in CD11b⁺ cells within the spleen of Bmal1^{-/-} mice when compared with Bmal1^{+/+} mice even though they had similar number of CD11b⁺ cells”. Are these mice the same mice maintained in a 12h:12h light:dark environment? There is no sufficient description of the mouse model. Are they OTII mice?

The authors assayed Opa1, Mfn1, Mfn2 and Fis1 mRNA expression but these genes did not display time-related variations. This is in contrast with previous observations. Time-course western blotting analysis of proteins encoded by Opa1, Mfn1, Mfn2 and Fis1 genes should be performed to confirm or exclude rhythmic patterns of expression.

A key player of mitochondrial dynamics and circadian clock function such as PGC1alpha, was not evaluated: please evaluate time-qualified expression and dynamics of PGC1alpha at all level in this experimental setting: expression level, challenge with inhibitors and

activators, PGC1alpha targets such as NRF1, NRF2, ERR- $\alpha/\beta/\gamma$.

TOM steady levels seem to exclude mitochondrial biogenesis...this is really strange. How the authors explain this result?

The authors suggest that the circadian changes of mitochondrial morphology must occur post-transcriptionally. Anyway, they not evaluate any post-transcriptional change (phosphatome, acetylome)

At least, the authors should evaluate miRNAome with customized and mito-directed miRNA-arrays

Figs. 3e,f: because not specified the experiment was likely performed in non-synchronized cells, if so it would be more informative to verify that the effect of oligomycin and FCCP was different at T12 and T24 and that the drug treatment decreased effectively the ATP level.

Figs. 4a,b staining of mitochondria with mitotracker appears quite different at different time-points, conflicting with the statement that the mitochondrial mass does not change following synchronization. The author should show the averaged probe fluorescence intensity/cell at the different time-points.

Fig. 5. The conclusion inferred by the authors that mitochondrial fusion elicits a more performing oxidative phosphorylation thereby enhancing antigen processing needs to be proved. The authors should show that under conditions inhibiting mitochondrial fission by Mdivi the mitochondrial activity effectively increases also at T24 post-synchronization.

Fig. 6a. Magnification of the confocal microscopy images showing representative cells are required to appreciate the specificity of the two calcium ion probes. Indeed, while Rhod-2 is known to accumulate electrophoretically into respiring mitochondria, Fluo-4 can permeate in its esterified form also the ER membrane. Thus, the author's conclusion about changes of the cytosolic calcium does not consider the contribution of the ER calcium. Given the tight association of mitochondria and ER in the form of contact points, the contribution of the ER to the redistribution of calcium should be assessed by testing the effect of inhibitors of the ER calcium channels (such as dantrolene vs BAPTA).

Fig.6d,e: as for point 5 the author should show the effect of the calcineurin inhibitor FK506 on respiration and by western blotting the impact of the treatment on the phosphorylation state of the mitochondrial fission factor Drp1.

Fig. 7a: the rhythmic expression of MCU needs to be supported by western blot analysis. It would be of interest to show if under condition of antigen processing (DQ-OVA assay) in DC the mitochondrial compartment undergoes some changes (such as polarization as shown for other immune-competent cells).

Finally the contribution of other authors that showed cell autonomous rhythmic changes of mitochondrial respiration as well as intracellular calcium redistribution should be acknowledged (Cela O. et al BBA 2016, 1863:596-606; Scrima R. et al. BBA 2020, 1867:118815).

Reviewer 1

We wish to thank the reviewers for the time taken to provide insightful and helpful suggestions. We hope that this point-by-point response will address the majority of their concerns. All changes made to the manuscript are highlighted in yellow.

Reviewer: *italic*
Authors: plain text

REVIEWER COMMENTS

Reviewer #1 (Remarks to the Author):

The manuscript “The circadian clock influences T cell responses to vaccination by regulating dendritic cell antigen processing through alterations in mitochondrial morphology and metabolism”, by Curtis et al., describes a mechanism by which calcium flux inside the cell (mitochondria/cytosol) can modulate the processing and presentation of antigens from DC to T cells in a circadian way, which could be the responsible for differential responses to vaccination. The study is interesting, noteworthy and well conducted. There are, however, a few concerns that demand specific explanations that could improve the manuscript.

We thank the reviewers for engaging positively with our study and we hope the responses below address the reviewers outstanding concerns.

Point 1) *Line 107-108: the sentence “As the transferred T cells were isolated from mice at the same circadian phase and the recipient mice were phase shifted as they were housed under shifted light schedules” is unclear. When the authors mention the isolation of transferred T cells, do they refer to their obtention from the original mice? Or is it the isolation from the experimental mice in order to analyze their proliferation?*

We fully accept that this wording was unclear and have now revised the text to the following in the manuscript:

Line 113 (manuscript)

We utilised an adoptive transfer model (**Fig. 1a**), where CD4⁺ T cells were isolated from OT-II mice on Day -1 at ZT3 and stained with cell trace violet (CTV). These CTV-stained OTII CD4⁺ T cells were immediately injected into recipient mice that had been phase shifted to either ZT7 or ZT19 in light cabinets to facilitate simultaneous experimentation. The next day ZT7 and ZT19 mice were immunised with OVA and whole cell pertussis (wcp) as an adjuvant, and 72 hours later (again corresponding to either ZT7 or ZT19), mediastinal lymph nodes were harvested from these mice to analyse proliferation and activation of the CTV-stained T cells. We reasoned that the approach of injecting the same CTV-stained OTII CD4⁺ T cells into phase-shifted recipient mice allowed us to more accurately interrogate the effect of the DC molecular clock on T cell activation (**Fig. 1a** and **Supplementary Fig. 1a**).

Point 2) On the other hand, the information of the specific lighting schedules, should be provided in the method section. It is unclear which mice are under the shifted lights schedule (those were injected at ZT7, or at ZT19, or both). It is also unclear which “circadian phase” is being mentioned.

We have now revised the materials and methods section to include the following:

Line 45 (M&M):

Adoptive transfer of CTV-labelled T cells, immunisations and isolation of mediastinal lymph nodes to conduct T cell characterisation by flow cytometry

For the adoptive transfer experiment (**Fig.1**), light cabinets were used to shift animals to specific light-dark cycles. Mice were given at least two weeks to entrain to any altered lighting schedule changes. This allowed us to transfer CTV-labelled OT-II T cells from donor mice taken at ZT3 into recipient mice who had been phase shifted to ZT7 or ZT19. This allowed simultaneous experimentation at ZT7 and ZT19. 24 hours later ZT7/ZT19 mice were immunised simultaneously with 1/50 human dose of a whole cell Pertussis (wcP) vaccine (Shan-5, Shantha Biotechnics Private, India) and OVA (EndoFit Ovalbumin, InvivoGen) 10 µg/mouse. Mice were immunised intraperitoneally (i.p.) with a final volume of 200 µl. 72 hours later, again corresponding to the respective phase of ZT7 and ZT19, mice were euthanised by CO₂ and mediastinal lymph nodes were harvested.

Furthermore, we have revised the schematic (**Fig.1a**) and have included it below to more clearly illustrate the lighting and experimental conditions. We hope both of these actions combined clarify the issue and also further address point 1.

Point 3) The legend to Figure 1 states that “Proliferation of CTV stained OTII CD4+ T cells harvested on Day 5 from mediastinal lymph node at ZT7 and ZT19”. In addition to immunization of mice at different times, were the tissues extracted at different times? This data is omitted in the experimental design. Indeed, in order to clarify the circadian organization of the experimental approach, a specific section should be added to the methods section.

We fully accept that the description of the *in vivo* adoptive transfer and immunisation experiments in terms of phase shifting and experimental procedure's and methods was

insufficient. This supports reviewers point 1 and point 2 in which we have revised the methods section (see above). We found a typographical error in that the tissues were extracted 3 days (72 hours later) and not 5 days as stated, which has now been corrected throughout.

Mice that were immunised at ZT7, their cells were extracted 72 hours later corresponding to ZT7 for that mouse. Similarly mice immunised at ZT19 their cells were extracted 72 hours later corresponding to ZT19. This has been made clearer in the methods but also in the schematic of Fig. 1a. which has been copied above.

We have also revised the Figure 1 legend to the following:

Figure 1. T cell activation and proliferation is dependent on time-of-day of immunisation

(a) Experimental design for adoptive transfer of labelled T cells and immunisations by circadian phase. CTV⁺ OT-II CD4⁺ T cells (harvested at ZT3) were transferred directly into ZT7 or ZT19 recipient mice. 24 hours later immunisations of ZT7 or ZT19 recipient mice occurred. Immunisations were performed using wcP vaccine + OVA 10 µg/mouse (n=6) or PBS control (n=3) and mediastinal lymph nodes harvested 72 hours later. (b and c) Proliferation of CTV⁺ stained OT-II CD4⁺ T cells harvested from mediastinal lymph node. (d) Percentage of divided and undivided CTV⁺ stained OT-II CD4⁺ T cells. (e) Representative plot of CD69⁺ expression on CTV⁺ stained OT-II CD4⁺ T cells. (f) Percentage of CD69⁺ expression on CTV⁺ stained OT-II CD4⁺ T cells. Data shown is mean with error bars representing ± SEM. Data were compared using two-tailed *t*-test, **p* < 0.05.

Point 4) Lines 114-116: *as shown in the Figure Suppl 1a, the analysis of CD69 was performed on the CTV-negative cells. The rationale for this should be better explained.*

This was a typographical error, the analysis was only conducted on the CTV-positive cells which were adoptively transferred into recipients. This error has been amended in the text.

Point 5) *Suppl. Fig. 1a-b: in the gating strategy used for lymph nodes, the presence of many CD8+CD4+ cells is striking since, theoretically, these cells mainly differentiate in the thymus. Additionally, the fluorescence signal of the CD8 marker is quite high, thus, for a better visualization of the cells that are positive for each marker, it is necessary to show the corresponding isotype controls.*

We thank the reviewer for highlighting this. Regarding the number of CD8+CD4+ cells, it is relevant to point out that this experiment measured the relatively small population of adoptively transferred CTV⁺-stained CD4⁺ T cells. Therefore we had to collect as many events as possible in order to have a statistically significantly CTV⁺ population. We have gone back and reanalysed all samples, and this double-positive population consists less than 0.56% of total CD3⁺ lymphocytes (see above). We also looked at CTV⁺ stained cells from total CD8 T cells together with double-positive and there was only a small contamination of CTV⁺ cells in this gate (0.02%). Therefore we are assured our results are not influenced by this. This double positive population could be an artifact, as we see a similar population in the spleens at similar frequency as we observe in the lymph node. As such we do not believe it is cross contamination from the thymus. It could be autofluorescence from red blood cells. For the manuscript, we have now redone the gating strategy for the manuscript with 80,000 events plotted to more clearly show the small percentage of this double positive population (**Supplementary Fig. 1a**).

We did not use isotype controls for CD8, because this population has high expression of the surface marker and separates nicely from other populations. Only OT-II CD4⁺ T cells were stained with CTV and all analysis is on CD4⁺ CTV⁺ cells.

Point 6) *Since some mechanisms are induced at ZT7 (Fig. 1d) and others at ZT19 (Fig. 2f, g), it is relevant to discuss this apparent contradiction. This appears to be relevant because the antigen processing is necessary for CD4 cells activation.*

The reviewer is correct in highlighting this. We believe the difference of the ZT effect can be explained as **Fig.1d** was administration of antigen in vivo via intraperitoneal injection (IP), whereas **Fig. 2f,g** was administration of antigen directly onto DCs ex vivo. Therefore, when we immunise animals by IP at ZT7, time is required for antigen to reach the DCs, and then additional time for sufficient uptake and processing to occur. Other reports suggest that this can take at least 6 hours.¹ Antigen uptake and processing will be much faster when antigen is applied directly to DCs ex vivo as in the case of **Fig. 2f,g (revised now to Fig. 2e,g and h)**. Therefore, our ex vivo data shows that the capacity for antigen processing is increasing from ZT7 to a maximum at ZT19, but from ZT19 to ZT7 it is decreasing (**Fig. 2e**). Therefore when you immunise mice at ZT7, DCs, taking into account the slower kinetics of uptake and processing in vivo, it follows that you see a higher response when mice are immunised at ZT7 versus ZT19.

To explain this further we have added the following description.

Line 370 (Manuscript)

Ex vivo analysis showed a steady increase in DC antigen processing from ZT7 to a peak at ZT19 and a decline. We rationalise that ZT7 injections align with a time that

allows optimal antigen processing capacity of DCs. Hence, immunising mice at ZT7 shows a higher response in comparison to ZT19.

Point 7) *The expression pattern of Nr1d1 is different in Suppl. Fig. 2 and 3. Suppl. Fig. 2 shows maximum levels 16-20 h post synchronization and minimal levels at 24 h (very similar to levels at 12h) post synchronization, but Suppl- Fig. 3 shows higher levels at 24 h than at 12 h post synchronization. Authors should revise these data.*

The differences between *Nr1d1* expression between Suppl. Fig. 2 and 3 are due to slight variations in the phase of this gene usually observed between different serum shock experiments. For clarity, we have included in **Supplementary Fig. 2a** the T12, T24 and T36 hours post serum shock data to show rhythmicity in *Bmal1*, *Per2* and *Nr1d1* expression and to show that *Bmal1* expression is antiphase to *Per2* and *Nr1d1*.

Point 8) *Lines 268-270: The result obtained with the mitochondrial fission inhibitor P110 (Sup. fig. 4a and b) should be further discussed since the data is somewhat contradictory.*

We appreciate that this was confusing in our original submission. Our rationale was to see if there might be differential effects of Mdivi-1 versus P110 which could be used to shed further light on mechanism. The differences between the two inhibitors are as follows, Mdivi-1 specifically inhibits DRP1 assembly and enzymatic function, whereas P110 blocks the DRP1/FIS1 interaction and DRP1 GTPase activity.²

There are a number of reports that Mdivi-1 modulates intracellular Ca^{2+} dynamics whereas P110 does not appear to affect Ca^{2+} .^{3,4}

We have since made the direct comparison between P110 and Mdivi-1 on antigen processing. We find that Mdivi-1 has a much greater effect on antigen processing than P110. This data has now been added to Supplementary Fig. 6 (see below), alongside the data where Mdivi-1 not P110 causes an increase in mitochondrial Ca^{2+} . Collectively this data indicates that both mitochondrial fusion and mitochondrial Ca^{2+} are required for enhanced antigen processing.

Supplementary Figure 6

We have included the following text

Line 293 (Manuscript)

We used a second molecule to increase mitochondrial fusion, a selective peptide P110 which promotes fusion by inhibiting DRP1 GTPase activity and the DRP1/FIS1 interaction.² We found that while P110 promotes mitochondrial fusion in *Bmal1*^{+/+}, it did not promote antigen processing to the same extent as Mdivi-1 (**Supplementary Fig. 6a**). Concurrently, we found that P110 did not increase mitochondrial calcium uptake (**Supplementary Fig. 6b**). Therefore, we reasoned that Mdivi-1 has such pronounced effects on antigen processing as it promotes both fusion and mitochondrial calcium uptake.

Point 9) Line 295-296: *It is confusing that the circadian analyses of mRNA levels shown in Fig. 7 was performed in a different way than the patterns shown in the remaining figures (particularly Suppl. Fig 2). Also, the explanation of the analyses performed with these data and the meaning of each color line in the figure is missing. In addition, 4 time points are not completely enough to calculate a circadian period. Finally, it is difficult to understand why its values are different from 24 hours since mice are housed under a standard light-dark schedule.*

We have now reanalysed the data in Fig. 7 using JTK_cycle which has become a significantly used method for measuring circadian rhythmicity in datasets.⁵ JTK analysis shows the period to be approximately 24 hours for all 4 genes. We have revised the graphs to show the expression of each mouse at each timepoint, instead of lines (**Fig. 7a**). We do appreciate that more time points are needed to assess true circadian rhythmicity, and we have included the following point as a limitations to our study.

Line 426 (Manuscript)

A number of aspects including exactly how the molecular clock regulates components of the MCU complex are **beyond the scope of this study and will require further investigation.**

An explanation of the circadian analysis has now been included in the materials and methods section

(Line 298)

MetaCycle was used to detect the presence of circadian rhythms in lumicycle data from synchronised PER2::Luciferase BMDCs and mRNA expression in splenocytes over the 24 hour cycle. For the PER2::Luciferase readings, normalized data was used. In MetaCycle, default settings and period length were set to 24 (h) for min and max period, cycMethod was set to “JTK” and circadian rhythms were identified with statistical significance $p < 0.05$.

Reviewer 1 Minor concerns

minor point 1) *Please refer to the methods (supplementary materials) in the main body of the manuscript.*

This has now been included

Line 340: Accompanying materials and methods are listed in supplementary section.

minor point 2) *Fig 1: it is unclear which statistical test that was used to analyze this data: when did the authors apply paired T-test and when Two-way ANOVA?*

We apologised for our error. The statistical test that was performed in both instances was unpaired *t*-test, and this has now been corrected.

minor point 3) *Fig 2: Is it unclear why pDC and macrophages are shown in the Suppl. Fig. 2, in particular because the response of total CD11c (and CD subtypes) and CD11b are shown in Fig. 2. I suggest including these graphs in Fig. 2.*

We agree with the reviewer. We have now combined all the bar graphs for cDC, cDC1, cDC2, migratory and resident DCs, plasmacytoid DC and macrophages into **Figure 2E-H** and moved all the dot plots into **Supplementary Fig. 3**.

minor point 4) *There are figures (like Fig 2c, 3e-h, 6d-e, 7b-d) that omit the time data. It is relevant to mention if this comparison compiled all time points or whether a single time point is shown.*

We thank the reviewer for pointing this out. We have now included either in the figure itself or in the figure legend the time point analysed or if it was performed under non-synchronised conditions. We also have created an additional schematic (see below) which is incorporated as **Fig. 2a**. This schematic depicts the correlation between hours post serum synchronisation and time-of-day. We believe this will further enhance understanding and clarity of the paper, allowing readers to quickly revert between hours post serum synchronisation results and the corresponding ZT.

minor point 5a) *Line 176 and Fig. 3: It is necessary to mention how the spare respiratory capacity is calculated. It could be explained in the legend of the figure or in the methods section and mentioned in the result section.*

A more comprehensive explanation of the MitoStress test methods and analysis is now included in Supplementary materials and methods (**Line 163**), and this test has been specifically mentioned in the results section of the manuscript (**Line 194**)

Minor point 5b) *Fig. 3e, f: the description of the result obtained with *Bmal1*^{-/-} cells is missing.*

We apologise for the omission. We have now included the following in the manuscript

Line 205 (manuscript)

Oligomycin and FCCP significantly reduced antigen processing in both *Bmal1*^{+/+} and *Bmal1*^{-/-} BMDCs (**Fig. 3e** and **3f**), indicating the importance of mitochondrial metabolism for antigen processing.

We have also added additional information into the Methods section on this assay.

Line 126 (M&M)

To investigate the impact of mitochondrial metabolism/function on antigen processing, BMDCs were pre-treated with oligomycin (10 ug/ml) and FCCP (100 nM) for 1 hour prior to performing the antigen processing assay and quantification by fluorescence on confocal microscopy.

minor point 6) Supplementary Materials:

a- Methods:

- “Flow cytometry for ex vivo analysis of splenic cells”, line 1: the paragraph also refers to lymph nodes.

This was a typographical error, as this section only refers to splenic cells, and has been corrected.

- “Data analyses”, line 8: it is confusing to refer to dexamethasone treatment for the synchronization step since all methods use serum shock for it. In the same line, the symbols (beta and gamma) are wrong.

This have now changed dexamethasone to serum shock, and revised the symbols.

- The explanation of migratory and resident DC identification is missing.

An explanation has now been included

Line 185 (manuscript)

The resident cDC are the major DC population in the spleen, and execute their antigen collection, processing and presentation within that lymphoid organ. The migratory cDCs constantly sample the tissue, and once activated, travel to the draining lymph node where they encounter naïve T cells.

- *Suppl. Fig 2: the terms Nr1d1 and Rev-erba are used for the same molecule, it is better to call it with the same nomenclature.*

We have now amended this and use of *Nr1d1* throughout

b- Figs. 2 and 6 and Suppl. Figs. 2 and 3: the title of the figures is missing.

The titles are now as follows

Fig. 2. Dendritic cells display robust rhythms in antigen processing

Fig. 6. Circadian rhythms in calcium localisation co-ordinate rhythms in antigen processing

Supplementary Fig. 2. Synchronised DCs produce robust rhythms in clock gene expression, but antigen uptake is not affected by the clock.

Supplementary Fig 3. Genes involved in mitochondrial morphology do not display circadian rhythms

- 1 Benson, R. A. *et al.* Antigen presentation kinetics control T cell/dendritic cell interactions and follicular helper T cell generation in vivo. *Elife* **4**, doi:10.7554/eLife.06994 (2015).

- 2 Qi, X., Qvit, N., Su, Y. C. & Mochly-Rosen, D. A novel Drp1 inhibitor diminishes aberrant mitochondrial fission and neurotoxicity. *J Cell Sci* **126**, 789-802, doi:10.1242/jcs.114439 (2013).
- 3 Cherubini, M., Lopez-Molina, L. & Gines, S. Mitochondrial fission in Huntington's disease mouse striatum disrupts ER-mitochondria contacts leading to disturbances in Ca(2+) efflux and Reactive Oxygen Species (ROS) homeostasis. *Neurobiol Dis* **136**, 104741, doi:10.1016/j.nbd.2020.104741 (2020).
- 4 Ruiz, A., Alberdi, E. & Matute, C. Mitochondrial Division Inhibitor 1 (mdivi-1) Protects Neurons against Excitotoxicity through the Modulation of Mitochondrial Function and Intracellular Ca(2+) Signaling. *Front Mol Neurosci* **11**, 3, doi:10.3389/fnmol.2018.00003 (2018).
- 5 Hughes, M. E., Hogenesch, J. B. & Kornacker, K. JTK_CYCLE: an efficient nonparametric algorithm for detecting rhythmic components in genome-scale data sets. *J Biol Rhythms* **25**, 372-380, doi:10.1177/0748730410379711 (2010).

Reviewer 2

We wish to thank the reviewers for the time taken to provide insightful and helpful suggestions. We hope that this point-by-point response will address the majority of their concerns. All changes made to the manuscript are highlighted in yellow.

Reviewer: *italic*

Authors: plain text

Reviewer #2 (Remarks to the Author):

The authors investigate the interplay between circadian rhythms and adaptive immunity focusing on the role of dendritic cells (DC) in this context. In the presented manuscript the authors show evidence of a causal linkage between the Bmal1-mediated rhythmic changes of the mitochondrial bioenergetic activity and the antigen processing in DC.The manuscript is well written, and the results obtained clearly commented and of interest however the following points are raised to which the authors are invited to reply.

We thank the reviewer for their time in reviewing the manuscript and for their considered response and positive remarks regarding our work. To aid with the review process, we have numerically labelled the reviewers comments as follows and all changes to the manuscript have been highlighted in yellow.

Point 1.) *The paper only partly contains new information. A recent article addressed the role of the biological clock in the dynamics of dendritic cells, even if in a different context, i.e. migration from the skin to lymph nodes, and should be cited and discussed in the light of the reported results (Holtkamp SJ, ...Scheiermann, C (2021). Circadian clocks guide dendritic cells into skin lymphatics. Nature immunology, 22(11), 1375–1381. <https://doi.org/10.1038/s41590-021-01040-x>)*

The reviewer is correct in pointing out this noteworthy publication, which was published *after* submission of our manuscript. However, the study by Holtkamp et al. addresses migration of DCs into the skin whereas ours addresses the role of DC's in promoting T cell responses to vaccination. Holtkamp et al. show that skin dendritic cells temporally migrate into the lymphatic vessels of the skin due to rhythmic gradients in the expression of the chemokines CCL21, LYVE-1, JAM-A and CD99 on skin lymphatic endothelial cells and CCR7, the DC receptor for CCL21. Thus the Holtkamp paper focuses solely on circadian control of cell migration and localisation of skin DCs. Their study does not cover any aspects of DC metabolism or antigen processing and presentation to T cells, therefore both manuscripts are complementary in terms of the importance of the DC circadian clock for vaccination but distinct in terms of mechanism and effect. We have included discussion of this paper in the revised manuscript.

Line 87 (manuscript) : Interestingly, Holtkamp et al. demonstrated that skin DCs preferentially migrate into lymphatic vessels during the mouse rest phase due to clock control of CCR7.

Line 373 (manuscript) : Our results are consistent with Holtkamp et al. who demonstrate that skin DCs preferentially migrate into lymphatic vessels at ZT7 due to rhythmic gradients in the chemokines CCL21, LYVE, JAM-A and CD99 on skin lymphatic endothelial cells and CCR7, the receptor on DCs for CCL21.

Point 2.) *The study design is not totally clear and sound and not all potential biases have been appropriately addressed. The authors mainly centre their study design on only a peculiar immunization methodology and a single not well described cellular model. Besides, the methods are not ever appropriate, not well described and not sufficient details are provided.*

We assume the reviewer is referring to the design of the in vivo study (**Fig. 1**). To address this, we have completely revised the description of the in vivo experiment both through revision of the schematic (**Fig.1a**), legend, results section and methods section (**also highlighted by reviewer 1 Major points 1-4**). We have also made significant clarifications of the methods section in response to queries from reviewer 1.

Point 3.) *Mice were maintained in a 12h:12h light:dark environment. To explore the genuine circadian nature of processes, mice should be housed in constant darkness to avoid masking effects. Please, explain the reasons that lead you to a different choice of housing schedule*

We decided to conduct the in vivo experiment in 12h:12h light:dark as we wished to show the impact of vaccine immunisation on DC responses in a light dark environment - which recapitulates the natural world. However, the reviewer is correct in stating that constant darkness could have been used to avoid possible masking effects in that instance. However we would argue that we have alternative data which sufficiently shows our effects to be circadian.

This includes (1) reduction of antigen processing observed ex vivo with *Bmal1* deletion in CD11b cells (**Fig. 2f**) and (2) the significant amount of in vitro data on BMDCs throughout the whole paper under synchronisation with and without *Bmal1* showing temporal and *Bmal1* dependent circadian control of antigen processing and that cellular metabolism underlies these effects. All in vitro experiments in synchronised DCs model the effect of the endogenous cellular clock.

Point 4.) *Please, describe in more detail how DCs were chosen/isolated/sorted, which kind of OVA peptide was used and how they were pre-loaded with OVA peptide in vitro. Why they did not use the OVA octameric peptide?*

To address the query on how DCs were chosen/isolated/sorted, we have revised the methods section to include the type of DC used in each type of experiment and how they were prepared. We have not used OVA peptides. Our study was focused on processing of OVA into antigens, therefore, full length ovalbumin protein was used in all assays in order to model antigen uptake, processing and presentation. Any type of OVA peptide (including octameric peptide) or loading of peptide onto MHC molecules would not be suitable for this study as it would bypass DC uptake and processing. Across the majority of experiments, we used DQ-OVA. DQ-OVA is a self-quenched conjugate of the full length ovalbumin protein which is strongly labelled with BODIPY dyes. As such it will exhibit bright fluorescence upon proteolytic degradation into singly, dye labelled peptides which can be detected either by flow cytometry or microscopy. This description of DQ-OVA has now been included.

Line 112 (M&M): DQ-OVA is a self-quenched labelled conjugate of the full-length ovalbumin protein, which produces a fluorescent green signal following cleavage into peptide fragments¹

Nonetheless, when addressing this comment, we realised there was a typographical error which did state OVA peptide, which likely caused this confusion – this has now been corrected to OVA protein and we apologise for this error.

Point 5.) “As the transferred T cells were isolated from mice at the same circadian phase and the recipient mice were phase shifted as they were housed under shifted light schedules. We reasoned that this approach allowed us to specifically measure the effect of the molecular clock in DCs on T cell activation” Please, clarify/correct/discuss in depth this statement, critical for your study design appropriateness

We have provided further details and clarity around the study design of the in vivo experiment as detailed in response to Reviewer 1 (**Major points 1-4**)

We also revised the following section in the discussion to enhance clarity.

Line 364 (manuscript) : We adoptively transferred CTV⁺ labelled OTII CD4⁺ T cells which were harvested from mice at one circadian phase and transferred into mice that were phase shifted to ZT7 or ZT19. Transfer of T cells from one circadian phase into either ZT7 or ZT19 mice allowed us to examine more specifically the DC circadian response in vivo.

Point 6.) *The authors should perform ex vivo re-challenge with OVA of mediastinal lymph node derived T cells to analyze cytokines secretion.*

Due to the demands of the in vivo experiment in terms of isolating and analysing T cell proliferation and activation by flow cytometry at exact ZT times, we prioritised the flow cytometry experiments on T cells. Therefore we could not perform ex vivo antigen re-challenge of the mediastinal lymph nodes to analyse cytokine secretion in our study. However, cytokine secretion is dependent on T cell proliferation and we would expect it to follow the same trends observed with T cell proliferation (**Fig. 1d**) and activation (**Fig. 1f**). Additionally, we have assessed cytokine production by T cell in in vitro experiments.

Our in vitro DC:T cell coculture experiments shows the following:

Lower IFN γ production in the absence of *Bmal1* in DCs (**Fig. 3h**)

Lower IFN γ production when we inhibit DC mitochondrial metabolism (**Fig. 3g**)

and lower IFN γ production when we inhibit uptake of mitochondrial calcium by inhibition of the mitochondrial calcium uniporter (**Fig. 7d**)

We believe that a combination of in vivo and in vitro assays allows us to demonstrate effects both on T cell proliferation and cytokine secretion, and collectively these results are consistent with our model (**Fig. 7E**).

Point 7.) *How PER2::Luc bone marrow derived dendritic cells (BMDCs) were produced ?*

The PER2::Luc BMDCs were isolated from PER2::Luc mice. This has been clarified further in the methods section.

Line 69 (M&M): This included *Bmal1LoxP/LoxP::Lyz2Cre* (*Bmal1*^{-/-}) which were compared with control *Lyz2Cre* (*Bmal1*^{+/+}). This also included bone marrow harvest from PERIOD2::luciferase mice which were used for lumicycle analysis (see below) and also C57Bl/6J mice as WT BMDCs

Point 8.) *Analysis of expression of the molecular clock genes, Period 2, Nr1d1 and Bmal1 in BMDCs by qPCR should be confirmed with western blotting of the same proteins*

We have included western blots for BMAL1 (**Supplementary Fig. 2b**). Commercially available antibodies for Period 2 and Nr1d1 are notoriously unreliable. After multiple attempts, we have been unable to obtain any reliable blots for Period 2 and Nr1d1 (see below) and as such are

unable to include any results in the paper. However, the most reliable indicator of molecular clock function is with a clock gene reporter such as PER2::Luciferase. This is shown in **Fig. 2c**, where we observe very high amplitude rhythms up to day 4-5 post serum synchronisation.

Point 9.) *The authors state “Bmal1 is the only non-redundant clock gene, and its deletion ablates all molecular clock function”. This is not correct...refer to <https://doi.org/10.1016/j.cub.2009.12.034> and <https://doi.org/10.1126/science.aaw7365>*

While there appears to be significant debate still about the findings from Ray et al., the data from Shi et al. do show that *Bmal2* can rescue a *Bmal1* knockout, when *ectopically* expressed. However, without this ectopic expression, *Bmal2* is downregulated when *Bmal1* is knocked out (Bunger et al.), making it unable to rescue a *Bmal1* knockout, which is the context we are referring to in our work and the reason that we are using this cell model. To clarify this distinction, we have reworded this line of the manuscript by modifying the text to say the following:

Line 153 (main manuscript): *Bmal1* is a core clock gene forming part of the positive arm. Deletion of *Bmal1* ablates the oscillation of the proteins in the negative arm of the core clock and cannot be compensated by any other paralogues in the native context.

Point 10.) *A better characterization at transcriptomic level should be performed, with scRNA-seq on Bmal1+/+ and Bmal1-/- CD11b+ cells, without and with OVA stimulation, with cell synchronization and followed by comprehensive bioinformatics analysis.*

A study such as this would be interesting in its own right. However, we would argue that a study such as this will not add anything further to our proposed model which is based on temporal changes in intracellular calcium and mitochondrial morphology. We assume the reviewer meant RNA-seq given the request is on CD11b+ cells? Also OVA protein will not significantly stimulate cells unless it is accompanied by an adjuvant and thus on its own we would not expect significant transcriptional changes. In order to properly assess circadian rhythmicity, an experiment would require samples to be taken every 2-3 hours for 48 hours. Both the running and analysis of such an experiment would take significant time and resource. In collaboration with Jennifer Hurley laboratory in Rensselaer, NY, we did perform such a study in synchronised bone marrow-derived macrophages (BMDMs) and analysed the transcriptome and proteome every 2 hours over 48 hours.² It is worth noting that this study took a full 5 year PhD to complete. However, in an effort to address the reviewers comment, we have mined this data transcriptomic and proteomic data. We could not detect any circadian

changes at the genes/proteins involved in antigen uptake, processing and presentation on MHCII providing further evidence that our mechanism is more dependent on non-genomic changes. We have now included a new schematic illustrating our model (**Fig. 7e**) which we believe will more easily illustrate our mechanistic model.

Point 11.) *Considering that chromatin regulators controlling rhythmicity manage the interplay between the biological clock and the immune system, epigenome remodeling evaluation with differential chromatin accessibility analysis should also be performed on these cell types (*Bmal1*^{+/+} and *Bmal1*^{-/-} CD11b⁺ cells)*

We believe that a study such as this would be of interest in its own right in order to understand circadian variation in chromatin accessibility in DCs, but would have minimal relevance to our model (**Fig. 7E**)

Point 12.) *There is an error in the labels in figure 2E, CD11b⁺ is repeated two times.*

We are not entirely clear as to what the reviewer is referring to here. Due to the inclusion of the new schematic as **Fig. 1a**, **Fig. 2e** is now **Fig. 2f**. CD11b⁺ is the first graph to denote that the % of CD11b⁺ cells is not different between *Bmal1*^{mye^{+/+}} and *Bmal1*^{mye^{-/-}} mice, however the next panel denotes the % of CD11b⁺ cells processing DQ-OVA between *Bmal1*^{mye^{+/+}} and *Bmal1*^{mye^{-/-}}. We have revised the axis labels to make this clearer. We also noted a typographical error in the legends, CD11b⁺ and CD11c⁺ as they were mislabelled as cDC11b⁺ and cDC11c⁺, and that has now been corrected.

Point 13.) *The authors use Mdivi-1, a quinazolinone derivative, as an inhibitor of dynamin-related protein 1 (DRP1), a crucial driver of mitochondrial fission, to inhibit Drp1-dependent mitochondrial fission and indirectly boost fusion-related patterns of mitochondrial respiration. Anyway, the results of this experimental protocol are absolutely expected, not particularly informative.*

We would respectfully argue that because the use of Mdivi-1 was important for validating our hypothesis, Our demonstration that Mdivi-1 can rescue deficits in antigen processing both due to time-of-day and *Bmal1* deficiency was an important finding. Therefore, it was crucial that we also demonstrated the impact of Mdivi-1 on enhancing mitochondrial fusion specifically in the DCs, and the positive correlation between fusion and antigen processing.

Point 14.) *The authors should take in account that in vivo DRP1 is phosphorylated with circadian rhythmicity influencing mitochondrial morphology and the biological clock mechanism. This series of effects could greatly lessen valuability of the results if DRP-1 inhibition is not performed at different time intervals. More informative could be the use of phosphates in a P-Drp1 rescue experiment.*

The reviewer is correct in stating that it would be important to investigate the effect of DRP1 inhibition at different time intervals. Indeed this is what we performed in **Fig. 5** in which we used Mdivi-1 (inhibitor of DRP1) and **Fig.6** where we used FK506, which via inhibition of the enzyme calcineurin will block dephosphorylation of serine 637 on DRP1. Both compounds were used at 12 hours post synchronisation (start of subjective day) and 24 hours post synchronisation (start of subjective night).

We appreciate that this may not have been clear in our original manuscript and have now included the hours post synchronisation above all panels and further description in the legend to provide more clarity to this point. We have also provided an additional schematic to clearly show the relationship between hours post synchronisation and time-of-day. (**Fig. 2a**)

We would argue that our results showing a statistically significant effect of Mdivi-1 (**Fig. 5c-d**) and FK506 (**Fig. 6f**) on antigen processing in *Bmal1*^{+/+} DCs at 24 hrs *but not at* 12 hrs post synchronisation provides further support that DRP1 is indeed endogenously clock controlled. At 12 hours post synchronisation the circadian clock has endogenously induced a more fused morphology through DRP1 inhibition. Thus, trying to further inhibit DRP1 with MDivi-1 or FK506 further has a nominal effect on antigen processing. In contrast at 24 hours post synchronisation DRP1 activity is high via the clock, as evidenced by *Bmal1*-dependent increased fission (**Fig. 4e**). Therefore inhibiting DRP1 at this circadian time with MDivi-1 or FK506 has a statistically significant effect on antigen processing. We would further add that our new data showing increased FIS1 expression specifically at 24 hours post synchronisation (**Supplementary Fig. 5c**) is another mechanism controlling DRP1 activity, given that FIS1 is key to DRP1 recruitment to the mitochondria.³ We hope that our inclusion of a final schematic (**Fig. 7e**) will more clearly demonstrate these molecular changes.

Point 15.) *The authors report the following statement “We also found that antigen processing was reduced in CD11b+ cells within the spleen of Bmal1-/- mice when compared with Bmal1+/+ mice even though they had similar number of CD11b+ cells”. Are these mice the same mice maintained in a 12h:12h light:dark environment? There is no sufficient description of the mouse model. Are they OTII mice?*

We appreciate how this confusion arose. We hope the following addresses this point

Firstly, all mice were maintained on a 12:12 schedule and this has not been included in the methods

Line 27 (M&M): All mice were maintained on as 12hr:12hr light:dark regimen with ad libitum food and water prior to experimentation.

This experiment was performed on transgenic mice in which *Bmal1* is excised in the myeloid lineage (*Bmal1*^{myeloid-/-}). This has been made clearer in the M&M section.

Line 19 (M&M): Mice with the gene *Bmal1* containing LoxP sites were kindly provided by Christopher Bradfield. *Bmal1LoxP/LoxP* were crossed with *Lyz2Cre* mice, which express Cre recombinase under the control of the *Lyz2* promoter to produce progeny that have *Bmal1* excised in the myeloid lineage. *Bmal1LoxP/LoxP::Lyz2Cre* (*Bmal1*^{myeloid-/-}) mice were compared with control *Lyz2Cre* (*Bmal1*^{myeloid+/+}). Offspring were genotyped to confirm the presence of LoxP sites and Cre recombinase.

Finally, when referring to these mice we have now revised figures and text from *Bmal1*^{+/+} and *Bmal1*^{-/-} to *Bmal1*^{myeloid+/+} and *Bmal1*^{myeloid-/-}

Line 172 (main manuscript) We also observed that antigen processing was reduced in CD11b⁺ cells within the spleen of *Bmal1*^{myeloid-/-} mice when compared with *Bmal1*^{myeloid+/+} mice even though they had similar number of CD11b⁺ cells.

Point 16.) *The authors assayed Opa1, Mfn1, Mfn2 and Fis1 mRNA expression but these genes did not display time-related variations. This is in contrast with previous observations. Time-course western blotting analysis of proteins encoded by Opa1, Mfn1, Mfn2 and Fis1 genes should be performed to confirm or exclude rhythmic patterns of expression.*

From previous reports, it appears that the rhythmicity of these proteins as either mRNA or protein level is dependent on the tissue of origin, with differences observed between liver, heart and muscle.⁴⁻⁸ Indeed in many of these reports there is a disconnect between cycling

at mRNA versus protein expression. Indeed, this observation agrees with our previous publication in which we observed only 15% of the circadian proteome had an oscillating mRNA in bone marrow-derived macrophages.² This implies that post-transcriptional control via that clock is a dominant mechanism controlling output, which is also in agreement with our findings.

As per the request of the reviewer we have performed time course western blot analysis of Opa1, Mfn1, Mfn2 and Fis1 genes. We found no rhythmicity in any of these genes at mRNA level (**supplementary Fig. 5a**). Interestingly, we did observe higher protein levels of Fis1 at 24 hours post synchronisation, versus 12 and 36 hours (**supplementary Fig. 5c**) and this correlates very nicely with our mitochondrial morphology results in which highest fission was observed at 24 hours post synchronisation (**Fig. 4e**). Fis1 protein expression was not rhythmic but was more highly expressed in cells lacking *Bmal1*^{-/-}, again in agreement with our morphology data (**Fig. 4e**). As discussed above (point 14), this is an important result because Fis1 is a key gene which recruits Drp1 to the mitochondria to mediate fission.³ Interestingly, when we mined the Hurley lab dataset² as discussed (**Point 10**), FIS1 was also cycling at the protein level but not at the mRNA level. We have described this in results section

(Line 243)

We also investigated mRNA and protein expression of key genes involved in mitochondrial morphology, such as *Fis1*, *Opa1*, *Mfn1* and *Mfn2*. None of these genes displayed any circadian variation at mRNA level (**Supplementary Fig. 5a**), however the mitochondrial fission gene *Fis1* was rhythmic at protein level in synchronised *Bmal1*^{+/+} but not *Bmal1*^{-/-} BMDCs with an increase at 24 hr post serum synchronisation (**Supplementary Fig. 5b-c**).

and discussion section (line 394)

Our data suggests that the molecular clock controls mitochondrial morphology by controlling levels and activity of the key fission proteins, Fis1 and Drp1.

Point 17.) *A key player of mitochondrial dynamics and circadian clock function such as PGC1alpha, was not evaluated: please evaluate time-qualified expression and dynamics of PGC1alpha at all level in this experimental setting: expression level, challenge with inhibitors and activators, PGC1alpha targets such as NRF1, NRF2, ERR-α/β/γ.*

The reviewer is correct in highlighting PGC-1alpha. The central role of PGC1alpha is mitochondrial biogenesis. We decided to directly assay mitochondrial biogenesis across time in our synchronisation model, with and without *Bmal1*. We believe this is more informative than assaying mRNA/protein expression or activity of PGC-1alpha and its targets. To achieve this we used mitotracker green (which measures mitochondrial mass) and quantified its fluorescence by flow cytometry. We did not observe any significant changes in mitochondrial mass either by time post synchronisation with presence/absence of *Bmal1*. This result has now replaced the TOM20 data (**Fig. 4h**).

Point 18.) *TOM steady levels seem to exclude mitochondrial biogenesis...this is really strange. How the authors explain this result?*

Given the additional experimentation we performed using mitotracker green (**Point 17**), we can now confidently exclude any circadian or *Bmal1* dependent effect on mitochondrial biogenesis. We have replaced the TOM20 data with the mitotracker green data, given the latter is a more definitive result.

Point 19.) *The authors suggest that the circadian changes of mitochondrial morphology must occur post-transcriptionally. Anyway, they not evaluate any post-transcriptional change (phosphatome, acetylome) At least, the authors should evaluate miRNAome with customized and mito-directed miRNA-arrays.*

We would argue that the main objective of our paper is to understand the connection between the DC molecular clock and antigen processing. To the end, we have made strong observations that the circadian changes in mitochondrial morphology modulate antigen processing in DCs. The exact mechanism by which the molecular clock directs circadian control of mitochondrial morphology is not the main focus. However, our new results on *Fis1* cycling at the protein level (**Point 16**) provide further evidence that the DCs clock controls mitochondrial morphology proteins at the post-transcriptional level and also our results with FK506 (**Point 14**) provide evidence that the clock is controlling mitochondrial morphology at the post translational level.

We would respectfully disagree with the reviewer that we should look at the miRNAome, given that miRNAs are not implicated in our model. Similar to reviewers **Point 10** and **11**, these omics experiments are interesting but they are *entirely standalone studies* due to the requirement for multiple time points for circadian analysis and thus require significant investment, time, expertise and resources.

Point 20.) *Figs. 3e,f: because not specified the experiment was likely performed in non-synchronized cells, if so it would be more informative to verify that the effect of oligomycin and FCCP was different at T12 and T24 and that the drug treatment decreased effectively the ATP level.*

We apologise for the confusion on this. **Fig. 3e** and **Fig. 3f** were performed under synchronised conditions at 12 hours post synchronisation. This has now been made clearer in the figure and legend. We chose 12 hours post synchronisation as this is the time when we see highest antigen processing (**Fig. 2c**) and mitochondrial fusion (**Fig. 4f**). We have performed the experiment investigating the effect of FCCP and oligomycin at 12 hours post synchronisation showing that both reduce ATP (**Supplementary Fig. 4a**). However as discussed in **Point 22**, we are not making the assumption that it is indeed higher ATP levels that is driving enhanced antigen processing, and have clarified it with this point.

Line 428 (manuscript): Future work should help to determine the mitochondrial metabolites that enhance antigen processing, and whether ATP is a contributing factor.

Point 21.) *Figs. 4a,b staining of mitochondria with mitotracker appears quite different at different time-points, conflicting with the statement that the mitochondrial mass does not change following synchronization. The author should show the averaged probe fluorescence intensity/cell at the different time-points.*

The reviewer is correct in their observation that the staining changes at different time points. The reason for this is that we have found that the optimal way to view mitochondrial morphology by confocal microscopy is using live cell imaging with MitoTracker Red. MitoTracker Red is a red-flourescent dye that stains mitochondria in live cells and its accumulation is dependent upon membrane potential. Hence that is why the staining changes at the different time points as we can infer that the membrane potential is changing in a circadian manner. This allowed us to quantify membrane potential (**Fig. 4g**) showing that it is antiphase to mitochondrial fission (**Fig. 4e**), and broadly in phase with elongated mitochondria (**Fig. 4f**). We have also added this additional line

Line 241 (Manuscript) : These circadian changes in mitochondrial potential were not due to changes in mitochondrial mass as analysis of mitochondrial mass using mitotracker green showed no time-of-day or *Bmal1* dependent variation (**Fig. 4h**).

Point 22.) *Fig. 5. The conclusion inferred by the authors that mitochondrial fusion elicits a more performing oxidative phosphorylation thereby enhancing antigen processing needs to be proved. The authors should show that under conditions inhibiting mitochondrial fission by Mdivi the mitochondrial activity effectively increases also at T24 post-synchronization.*

We decided that the best approach to investigate this point, was to test if we could enhance ATP levels at 24 hours post synchronisation with both compounds Mdivi-1 and FK506. However we did not see any alterations in ATP levels with either compound (**Supplementary Fig. 4b**) even though we observe an increase in mitochondrial fusion, mitochondrial calcium and antigen processing with these compounds. Although we had not directly proposed to know the mitochondrial factor connecting mitochondrial metabolism to antigen processing, we agree with the reviewer that it could have been inferred to be ATP release through OXPHOS. We have revised the whole manuscript where we have replaced wording such as “ATP” and “energy production” to now include the phrase “mitochondrial metabolism”. We have made it clearer in both the results section and the schematic (**Fig. 7e**) that further studies are required to understand what is directly linking the alterations in mitochondrial metabolism to antigen processing.

Line 310 (Manuscript) : We hypothesised that the circadian control of antigen processing via mitochondria could be through ATP (**Fig. 3d**). However, neither Mdivi-1 or FK506 was able to enhance ATP levels (**Supplementary Fig. 4b**).

Point 23.) *Fig. 6a. Magnification of the confocal microscopy images showing representative cells are required to appreciate the specificity of the two calcium ion probes. Indeed, while Rhod-2 is known to accumulate electrophoretically into respiring mitochondria, Fluo-4 can permeate in its esterified form also the ER membrane. Thus, the author’s conclusion about changes of the cytosolic calcium does not consider the contribution of the ER calcium. Given the tight association of mitochondria and ER in the form of contact points, the contribution of the ER to the redistribution of calcium should be assessed by testing the effect of inhibitors of the ER calcium channels (such as dantrolene vs BAPTA).*

We would respectfully argue that both Rhod-2 and Fluo-4 are well established Ca⁺⁺ -sensitive probes with significant specificity for the mitochondria versus cytoplasm.^{9,10} However, we do appreciate the reviewer’s comments regarding the impact of ER calcium and as such we did examine the effect of antigen processing with dantrolene vs BAPTA. Although we observed a very slight decrease in antigen processing with BAPTA (10 μ M) and a very slight increase with Dantrolene (20 μ M) these effects were minimal in comparison to Mdivi-1 (see below).

As such we don't believe these preliminary results warrant inclusion in the paper, but will be the subject of future work where we will specifically use the ER calcium dye (MAG-Fluo-4), and we have made reference to this in the manuscript (also included in **Point 26**).

Line 428 (manuscript): The importance of the ER in terms of mitochondrial Ca^{2+} efflux and antigen processing will also be the subject of future studies

Point 24.) *Fig.6d,e: as for point 5 the author should show the effect of the calcineurin inhibitor FK506 on respiration and by western blotting the impact of the treatment on the phosphorylation state of the mitochondrial fission factor Drp1.*

As discussed (**Point 22**), we did not observe an increase in ATP levels with FK506, prompting us to state in the revised manuscript that the role of metabolite(s) connecting the circadian changes in mitochondrial morphology and metabolism with antigen processing requires further investigation.

We have now determined by western blotting that FK506 treatment can increase phosphorylation of Drp1-Ser637 (**Supplementary Fig. 7**) in agreement with others.¹¹ This will increase mitochondrial fusion by preventing Drp1 recruitment to the mitochondria and is in agreement with our results showing that FK506 increases mitochondrial fusion (**Fig. 6e**).

Point 25). *Fig. 7a: the rhythmic expression of MCU needs to be supported by western blot analysis.*

It would be of interest to show if under condition of antigen processing (DQ-OVA assay) in DC the mitochondrial compartment undergoes some changes (such as polarization as shown for other immune-competent cells).

Western blot analysis for component of the MCU complex is not possible due to lack of specific antibodies.

For the second point, this would be interesting but not directly relevant to our mechanism. Such an experiment is technically challenging given that both the emission spectrum from DQ-OVA and MitoTracker Red are overlapping.

Point 26.) *Finally the contribution of other authors that showed cell autonomous rhythmic changes of mitochondrial respiration as well as intracellular calcium redistribution should be acknowledged (Cela O. et al BBA 2016, 1863:596-606; Scrima R. et al. BBA 2020, 1867:118815).*

We sincerely apologise for the omission of these highly relevant papers.

Cela O. et al. and Scrima R. et al. has now been included in the introduction section along with Schmitt et al. Cell Metabolism 2018

Line 72: These oscillations in oxidative phosphorylation and ATP generation have been demonstrated across a range of cell types, and are associated with the NAMPT-NAD-SIRT1/3 axis and dependent on circadian modification of DRP1 phosphorylation.

Scrima R. et al. has been included in the discussion section

Line 406: Synchronised HepG2 cell cultures display a circadian rhythm in mitochondrial Ca²⁺ which drives rhythmic mitochondrial respiration &

Line 428: The importance of the ER in terms of mitochondrial Ca²⁺ efflux and antigen processing will also be the subject of future studies

- 1 Daro, E. *et al.* Polyethylene glycol-modified GM-CSF expands CD11b(high)CD11c(high) but not CD11b(low)CD11c(high) murine dendritic cells in vivo: a comparative analysis with Flt3 ligand. *J Immunol* **165**, 49-58, doi:10.4049/jimmunol.165.1.49 (2000).
- 2 Collins, E. J. *et al.* Post-transcriptional circadian regulation in macrophages organizes temporally distinct immunometabolic states. *Genome Res*, doi:10.1101/gr.263814.120 (2021).
- 3 Loson, O. C., Song, Z., Chen, H. & Chan, D. C. Fis1, Mff, MiD49, and MiD51 mediate Drp1 recruitment in mitochondrial fission. *Mol Biol Cell* **24**, 659-667, doi:10.1091/mbc.E12-10-0721 (2013).
- 4 Gong, C. *et al.* The daily rhythms of mitochondrial gene expression and oxidative stress regulation are altered by aging in the mouse liver. *Chronobiology international* **32**, 1254-1263, doi:10.3109/07420528.2015.1085388 (2015).
- 5 Jacobi, D. *et al.* Hepatic Bmal1 Regulates Rhythmic Mitochondrial Dynamics and Promotes Metabolic Fitness. *Cell Metab* **22**, 709-720, doi:10.1016/j.cmet.2015.08.006 (2015).
- 6 Kohsaka, A. *et al.* High-fat diet disrupts behavioral and molecular circadian rhythms in mice. *Cell metabolism* **6**, 414-421, doi:10.1016/j.cmet.2007.09.006 (2007).
- 7 van Moorsel, D. *et al.* Demonstration of a day-night rhythm in human skeletal muscle oxidative capacity. *Mol Metab* **5**, 635-645, doi:10.1016/j.molmet.2016.06.012 (2016).
- 8 Wefers, J. *et al.* Day-night rhythm of skeletal muscle metabolism is disturbed in older, metabolically compromised individuals. *Mol Metab* **41**, 101050, doi:10.1016/j.molmet.2020.101050 (2020).
- 9 Mirnikjoo, B., Balasubramanian, K. & Schroit, A. J. Mobilization of lysosomal calcium regulates the externalization of phosphatidylserine during apoptosis. *J Biol Chem* **284**, 6918-6923, doi:10.1074/jbc.M805288200 (2009).

- 10 Wang, H. *et al.* Redistribution of subcellular calcium and its effect on apoptosis in primary cultures of rat proximal tubular cells exposed to lead. *Toxicology* **333**, 137-146, doi:10.1016/j.tox.2015.04.015 (2015).
- 11 Schmitt, K. *et al.* Circadian Control of DRP1 Activity Regulates Mitochondrial Dynamics and Bioenergetics. *Cell Metab* **27**, 657-666 e655, doi:10.1016/j.cmet.2018.01.011 (2018).

REVIEWERS' COMMENTS

Reviewer #1 (Remarks to the Author):

I have carefully revised the new manuscript and the replies to both reviewers and acknowledge that the authors have made a fine work in their reanalyses and rewriting of the text, so I recommend publication at this point.

Reviewer #2 (Remarks to the Author):

The authors have adequately responded to the corrections and suggestions provided by the referee.

I appreciate the arguments in support of their decisions regarding some controversies concerning differences of interpretation